# Maternal preconceptional and prenatal exposure to El Niño Southern Oscillation levels and child mortality: a multi-country study

Hongbing Xu[1,2], Castiel Chen Zhuang ®[3] ✉, Vanessa M. Oddo[4], Espoir Bwenge Malembaka ®[5,6,7], Xinghou He[1,2], Qinghong Zhang ®[8] & Wei Huang[1,2]

El Niño Southern Oscillation (ENSO) has been shown to relate to the epidemiology of childhood infectious diseases, but evidence for whether they increase child deaths is limited. Here, we investigate the impact of mothers' ENSO exposure during and prior to delivery on child mortality by constructing a retrospective cohort study in 38 low- and middle-income countries. We find that high levels of ENSO indices cumulated over 0–12 lagged months before delivery are associated with significant increases in risks of under-five mortality; with the hazard ratio ranging from 1.33 (95% confidence interval [CI], 1.26, 1.40) to 1.89 (95% CI, 1.78, 2.00). Child mortality risks are particularly related to maternal exposure to El Niño-like conditions in the 0th–1st and 6th–12th lagged months. The El Niño effects are larger in rural populations and those with unsafe sources of drinking water and less education. Thus, preventive interventions are particularly warranted for the socio-economically disadvantaged.

Substantial declines in under-five mortality (U5M) have been achieved globally in the past decades, which are partially attributable to the expanded provision of various life-saving health interventions and rising levels of individual education and income, especially in reproductive-age women[1,2]. U5M is widely recognized as the most crucial and sensitive indicator of health and socioeconomic status[3]. Despite recorded improvements in child survival in numerous regions, recent work suggested that deaths in children under 5 years of age numbered 5.9 million worldwide in 2015, and ≥80% of the cases were from low- and middle-income countries (LMICs), such as Sub-Saharan Africa and Asia[4]. To improve child survival, the Sustainable Development Goals have recently proposed a global target for all countries to end preventable deaths in under-five children by 2030[5]. Therefore, identifying potential risk factors is particularly essential for reducing U5M.

Research into the El Niño Southern Oscillation (ENSO) events has raised concerns about the large-scale detrimental effects of climate variability on human beings, particularly among pregnant women, infants, and children[6–8]. ENSO is a periodic and irregular climatic phenomenon of variations in sea surface temperature, pressure, and wind in the equatorial Pacific Ocean that occurs every 3–7 years and acts as the crucial driver of global climate variability[9]. Two opposing

[1]Department of Occupational and Environmental Health, Peking University School of Public Health, Beijing, China. [2]Peking University Institute of Environmental Medicine, Beijing, China. [3]Peking University School of Economics, Beijing, China. [4]Department of Kinesiology and Nutrition, College of Applied Health Sciences, University of Illinois Chicago, Chicago, IL, USA. [5]Center for Tropical Diseases and Global Health, Université Catholique de Bukavu, Bukavu, Democratic Republic of the Congo. [6]Faculty of Medicine, Université Catholique de Bukavu, Bukavu, Democratic Republic of the Congo. [7]Department of Epidemiology, Johns Hopkins Bloomberg School of Public Health, Baltimore, MD, USA. [8]Department of Atmospheric and Oceanic Sciences, School of Physics, Peking University, Beijing, China. ✉e-mail: zogcee@gmail.com

climate patterns of ENSO, including warm (El Niño-like) and cold (La Niña-like) conditions, can heighten the probability of extreme weather events in certain regions, which might trigger a wide array of health risks[10]. An episode of El Niño or La Niña generally lasts 9–12 months, although it can sometimes last for years[11]. It has been shown that the magnitude of the impacts attributable to ENSO varies across regions, depending on how intensely El Niño/La Niña-like conditions affect local weather patterns (e.g., heat waves, cold spells, and droughts) of an area (namely, tele-connected impacts)[12]. El Niño events are observed in association with increased sea surface temperatures in the western Indian Ocean, above-normal rainfall in Eastern Africa, and above-normal temperatures in parts of Southeast Asia[13]. Pregnancy is perceived as a physiological "stress test" because the maternal body is challenged with a variety of adaptive changes in cardiorespiratory, endocrine, and immune function[14]. These maternal responses, along with the dynamic alterations in between fetoplacental circulation and rapid growth of fetus, may confer both mothers and children more susceptible to the adverse effects of climate-related stresses[14]. Thus, it is plausible that ENSO anomalies may worsen maternal and fetal health, thereby leading to a heightened risk of child deaths. So far, higher mortality risks have been linked to extreme levels of ENSO, such as El Niño-like conditions among older adults, indicating that ENSO may be a crucial environmental determinant of human health[15,16]. However, previous research on mortality risks associated with ENSO exposure primarily focused on the elderly and was typically conducted within a single country. Although emerging evidence suggests that El Niño-like conditions could contribute to outbreaks of water-borne diseases and underweight status in children[17,18], the evidence for other aspects of child health (e.g., survival status) is limited. To the best of our knowledge, the association between ENSO and child mortality has not been rigorously investigated. A recent report from the World Health Organization (WHO) indicates that U5M is one of the useful indicators for monitoring the overall impacts of ENSO events including El Niño[11]. Further, the effects of climate-related stresses may further exacerbate the intergenerational cycle of malnutrition[11,19]. Given that El Niño episodes have developed in 2023 and will likely occur more frequently in the near future, and with El Niño cycles continuing to take place against a background of climate change in general, it is likely that the stronger El Niño events can be associated with even more extreme weather variables in the future, and it is important to determine the impact of mothers' preconceptional and prenatal exposure to ENSO levels on child survival and discern sensitive windows of exposure, which could have critical clinical implications for the mitigation of U5M.

To this end, we hypothesized that climate anomalies driven by ENSO would heighten child mortality risks in a relatively large geographic area that has often been regarded as tele-connected to ENSO. In this paper, we constructed a retrospective cohort of children under 5 years of age by pooling all available mortality data from 160 nationally representative Demographic and Health Surveys (DHS). Our main objective was to characterize the relationship between maternal ENSO exposure during and prior to delivery and U5M for multiple LMICs and to identify potentially critical exposure windows. We also examined the effect modification of indicators relevant to socioeconomic status, quality of health services, as well as environmental conditions that may confer susceptibility of climate-related effects, and to estimate potential mortality burden attributable to extreme levels of ENSO.

## Results

### Study participants and ENSO exposure
As shown in Fig. 1A, 34 of 38 LMICs (29 of 31 African and 5 of 7 Asian countries) with all data for crucial covariates available were included the primary analysis. The sample size by country is summarized in Supplementary Tables S1 and S2. Table 1 and Supplementary Table S3

summarize the descriptive statistics of children aged under 5 years and their mothers. As shown in Table 1, a total of 1499,727 children aged 0–5 years from 34 LMICs were included in the primary analysis, of which 103,557 children died over 39,987,500 person-months of follow-up. Among these deaths, 45,450 neonatal deaths occurred within the first month after birth, and 80,048 infant deaths occurred in the first year of their life. Of all the observations in the sample, nearly 47.0% had mothers younger than 25 years old, 70.2% lived in rural areas, 68.8% had mothers with a primary educational level or lower, and 62.4% did not have safe sources of drinking water. Supplementary Fig. S1 displays substantial variations in ENSO measures, including the multivariate El Niño index (MEI) and the oceanic Niño index (ONI), over the past decades. Overall, month-to-month change patterns were similar across all ENSO measures, with high values observed during 1982–1984, 1997–1998, and 2015–2016, indicating the impacts of El Niño events. Results from Spearman correlation analysis showed that ENSO measures were related to local weather conditions in different ways (Fig. 1B, C). For instance, higher MEI levels were negatively correlated with precipitation in the southern regions of India, but positive correlations were found with temperature, suggesting that warmer conditions posed by ENSO could shift precipitation and temperature patterns and thereby lead to this region getting drier and hotter.

### Primary regression results
Figure 2 presents the overall cumulative associations between maternal ENSO exposure levels up to 12 months prior to delivery and child survival (neonatal mortality [death at <1 month of age], infant mortality [death at <12 months of age], and all U5M [death at <60 months of age]). In general, these curves showed an approximately J-shaped relationship between ENSO and all-cause mortality in children, and higher ENSO levels (El Niño-like conditions) were associated with greater mortality risks than lower ENSO levels (La Niña-like conditions). The exposure–response curve was applied to quantify the effect estimates, and the value (zero) reflecting neutral ENSO condition was used as the reference (see the "Statistical analysis" subsection under the "Methods" section). Comparing the 90th percentile (P90) (2.0) level of MEI exposure to the reference value (zero) at lag 0–12 months, the hazard ratio (HR), i.e., the risk at the specified level of exposure divided by the risk at the reference level, of neonatal, infant, and U5M was 1.18 (95% confidence interval [CI], 1.12, 1.25), 1.37 (95% CI, 1.31, 1.43), and 1.48 (95% CI, 1.43, 1.54), respectively. Relative to the reference value (zero), the HR of neonatal mortality ranged from 0.82 (95% CI, 0.77, 0.87) to 0.96 (95% CI, 0.93, 0.99) for ENSO precipitation index (ESPI), ONI, and Niño 1 + 2 at their 10th percentiles (P10), whereas the associations were stronger at the P90 of exposure, with effect estimates ranging from 1.31 (95% CI, 1.23, 1.39) to 1.67 (95% CI, 1.56, 1.78). Exposure to higher Niño 3.4 levels before birth was also associated with increased risks of child mortality.

Figure 3 graphically depicts the lag association pattern of ENSO exposure at individual lagged months prior to the birth of each child using contour plots. Overall, we found that extremely high levels of ENSO measures in multiple periods, and particularly the 0th–1st and 6th–12th lagged months were associated with an increased risk of child mortality. We showed that the maximum HR of infant and U5M for MEI at the current (i.e., 0th lagged) month was 1.09 (95% CI, 1.05, 1.13) and 1.11 (95% CI, 1.07, 1.15), respectively. For ESPI, higher risks of U5M were found in association with various consecutive (e.g., the 0th–1st and 6th–12th) lagged months of exposure, with effect estimates ranging between 1.02 (95% CI, 1.01, 1.03) and 1.09 (95% CI, 1.06, 1.12). Similar association patterns were also found for other ENSO indices, such as ONI, Niño 1 + 2, and Niño 3.4.

### Exploratory analyses
Figure 4, Supplementary Figs. S2–S6, and Supplementary Table S4 present the associations of child survival with maternal ENSO exposure

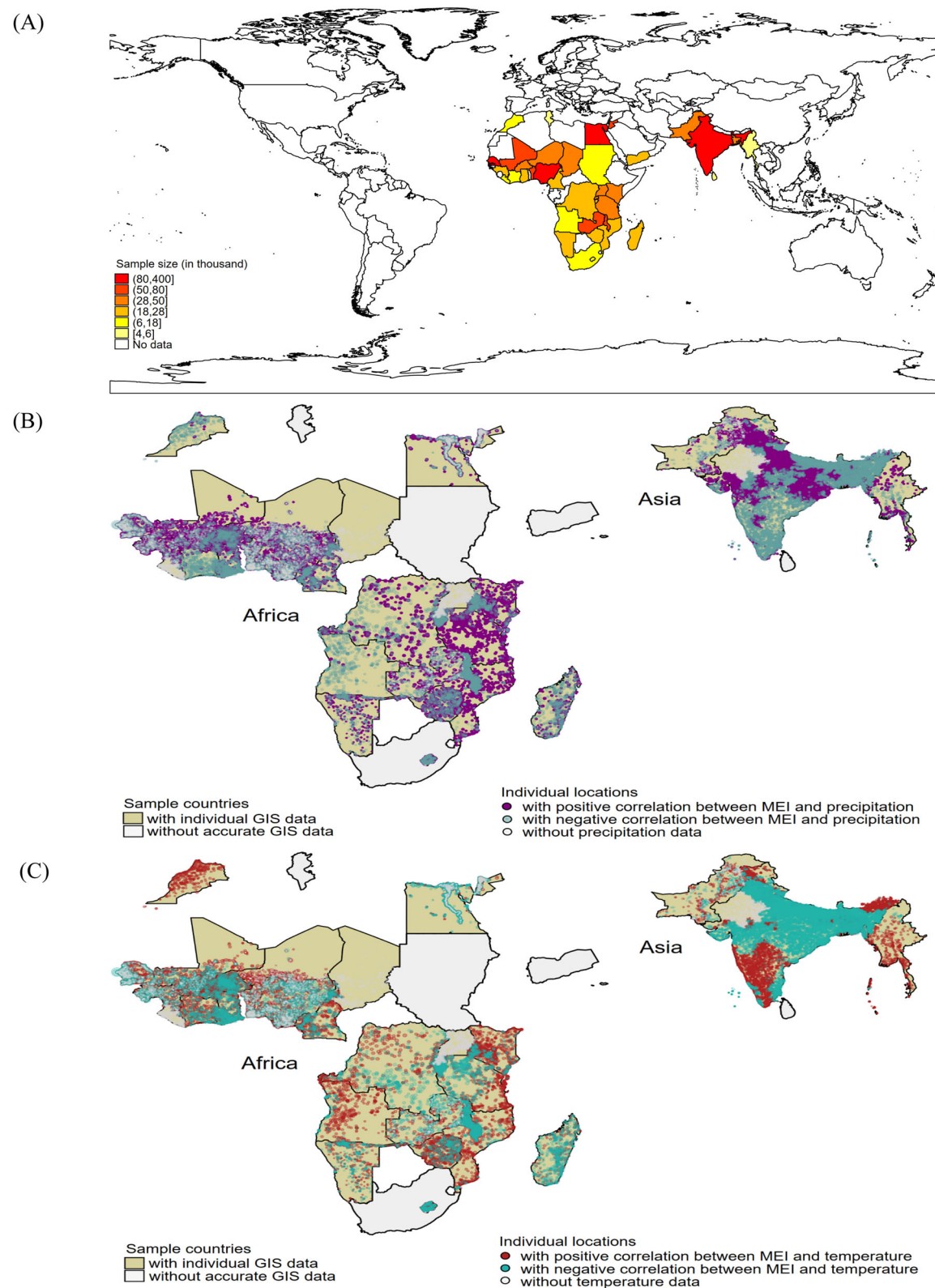

**Fig. 1 | Distribution of the study areas with ENSO tele-connected impacts.**
**A** Historical (i.e., based on survey time) geographic locations of 38 countries with
data on under-five mortality included in this analysis; (**B**) and (**C**) are monthly
correlations of 2-month lag of MEI and meteorologic parameters (precipitation and
temperature respectively) within a 10-km circular buffer around each DHS cluster.
ENSO El Niño Southern Oscillation, MEI multivariate El Niño index.

prior to delivery stratified by participants' characteristics. We found
suggestive heterogeneity of the associations between El Niño-like con-
ditions and mortality risks across a suite of child-level, mother-level, and
household-level covariates, including urbanicity, safe water access,

maternal education, and geographic zone (Fig. 4 and Supplementary
Figs. S2–S6). Comparing to the reference value, the HR of infant mor-
tality for MEI at the P90 of exposure was 1.22 (95% CI, 1.12, 1.34) in
mothers from urban areas, but the effect in those who resided in rural

**Table 1 | Characteristics of children under 5 years of age and their mothers based on the analyzed sample**

| Characteristics | Entire cohort | All-cause child mortality | | |
|---|---|---|---|---|
| | | Neonatal (<1 month of age) | Infant (<12 months of age) | Under 5 (<60 months of age) |
| Children, n | 1,499,727 | 45,450 | 80,048 | 103,557 |
| Total follow-up in month, n | 39,987,500 | 0 | 172,750 | 661,046 |
| Maternal covariates | | | | |
| Mother age at childbirth in years, mean (SD) | 26.5 (6.4) | 26.2 (7.0) | 26.3 (6.9) | 26.4 (6.9) |
| Mother age group in years, n (%) | | | | |
| [15, 19] | 229,054 (15.3) | 9532 (21.0) | 16,145 (20.2) | 20,352 (19.7) |
| [20, 24] | 475,918 (31.7) | 13,649 (30.0) | 23,644 (29.5) | 30,479 (29.4) |
| [25, 29] | 386,840 (25.8) | 9672 (21.3) | 17,810 (22.2) | 23,382 (22.6) |
| [30, 34] | 232,555 (15.5) | 6548 (14.4) | 11,791 (14.7) | 15,466 (14.9) |
| ≥35 | 175,360 (11.7) | 6049 (13.3) | 10,658 (13.3) | 13,878 (13.4) |
| Urbanicity, n (%) | | | | |
| Urban | 446,986 (29.8) | 11,623 (25.6) | 19,501 (24.4) | 24,367 (23.5) |
| Rural | 1,052,741 (70.2) | 33,827 (74.4) | 60,547 (75.6) | 79,190 (76.5) |
| Marital status, n (%) | | | | |
| In marriage | 1,389,805 (92.7) | 42,066 (92.6) | 73,192 (91.4) | 94,507 (91.3) |
| Others | 109,922 (7.3) | 3384 (7.4) | 6856 (8.6) | 9050 (8.7) |
| Education, n (%) | | | | |
| No education | 615,339 (41.0) | 21,565 (47.4) | 39,347 (49.2) | 53,113 (51.3) |
| Primary | 416,398 (27.8) | 12,764 (28.1) | 23,852 (29.8) | 30,977 (29.9) |
| Secondary or higher | 467,990 (31.2) | 11,121 (24.5) | 16,849 (21.0) | 19,467 (18.8) |
| Wealth quantile of household, n (%) | | | | |
| Poorest or poorer—bottom 40% | 681,749 (45.5) | 22,659 (49.9) | 40,589 (50.7) | 53,301 (51.5) |
| Middle or richer—middle 40% | 575,645 (38.4) | 17,038 (37.5) | 29,791 (37.2) | 38,277 (37.0) |
| Richest—top 20% | 242,333 (16.2) | 5753 (12.7) | 9668 (12.1) | 11,979 (11.6) |
| Household access to safe water, n (%) | | | | |
| Yes | 564,197 (37.6) | 14,045 (30.9) | 23,517 (29.4) | 28,889 (27.9) |
| No | 935,530 (62.4) | 31,405 (69.1) | 56,531 (70.6) | 74,668 (72.1) |
| Household access to toilet facilities, n (%) | | | | |
| Yes | 490,686 (32.7) | 17,455 (38.4) | 30,648 (38.3) | 39,819 (38.5) |
| No | 1,009,041 (67.3) | 27,995 (61.6) | 49,400 (61.7) | 63,738 (61.5) |
| Child covariates | | | | |
| Sex, n (%) | | | | |
| Male | 765,172 (51.0) | 26,045 (57.3) | 43,814 (54.7) | 55,629 (53.7) |
| Female | 734,555 (49.0) | 19,405 (42.7) | 36,234 (45.3) | 47,928 (46.3) |
| Birth order, n (%) | | | | |
| First | 377,152 (25.1) | 13,541 (29.8) | 21,205 (26.5) | 25,555 (24.7) |
| Second | 328,296 (21.9) | 8371 (18.4) | 14,721 (18.4) | 18,900 (18.3) |
| Third or fourth | 412,878 (27.5) | 10,586 (23.3) | 19,849 (24.8) | 26,435 (25.5) |
| ≥Fifth | 381,401 (25.4) | 12,952 (28.5) | 24,273 (30.3) | 32,667 (31.5) |
| Child delivery place, n (%) | | | | |
| Institution | 854,481 (57.0) | 23,480 (51.7) | 38,781 (48.4) | 47,971 (46.3) |
| Home | 645,246 (43.0) | 21,970 (48.3) | 41,267 (51.7) | 55,586 (53.7) |
| Geographic zone, n (%) | | | | |
| Northern and Western Africa | 481,755 (32.1) | 15,282 (33.6) | 27,999 (35.0) | 39,243 (37.9) |
| Central, Eastern, and Southern Africa | 490,592 (32.7) | 13,772 (30.3) | 28,379 (35.5) | 37,763 (36.5) |
| South, Southeast, and Western Asia | 527,380 (35.2) | 16,396 (36.1) | 23,670 (29.6) | 26,551 (25.6) |

The numbers in the parentheses can be an SD or %, while the numbers outside the parentheses can be a mean or count, as denoted above.

n count or the number of observations, SD standard deviation, % percentage point.

areas was higher (HR, 1.41; 95% CI, 1.35, 1.48). Furthermore, unsafe sources of drinking water and lower maternal education could exacerbate the detrimental effects on children. There were no consistent results of effect modification by birth quarter across different ENSO indices. For effect modification by geographic zone, the stronger impacts of ENSO on child survival occurred in Africa, especially individuals in the central, eastern, and southern parts of Africa. Based on available data for the type of cook fuels, the deleterious effect of ENSO on U5M was lower for those who were from households using clean fuels for cooking activities (Supplementary Table S4). Additionally,

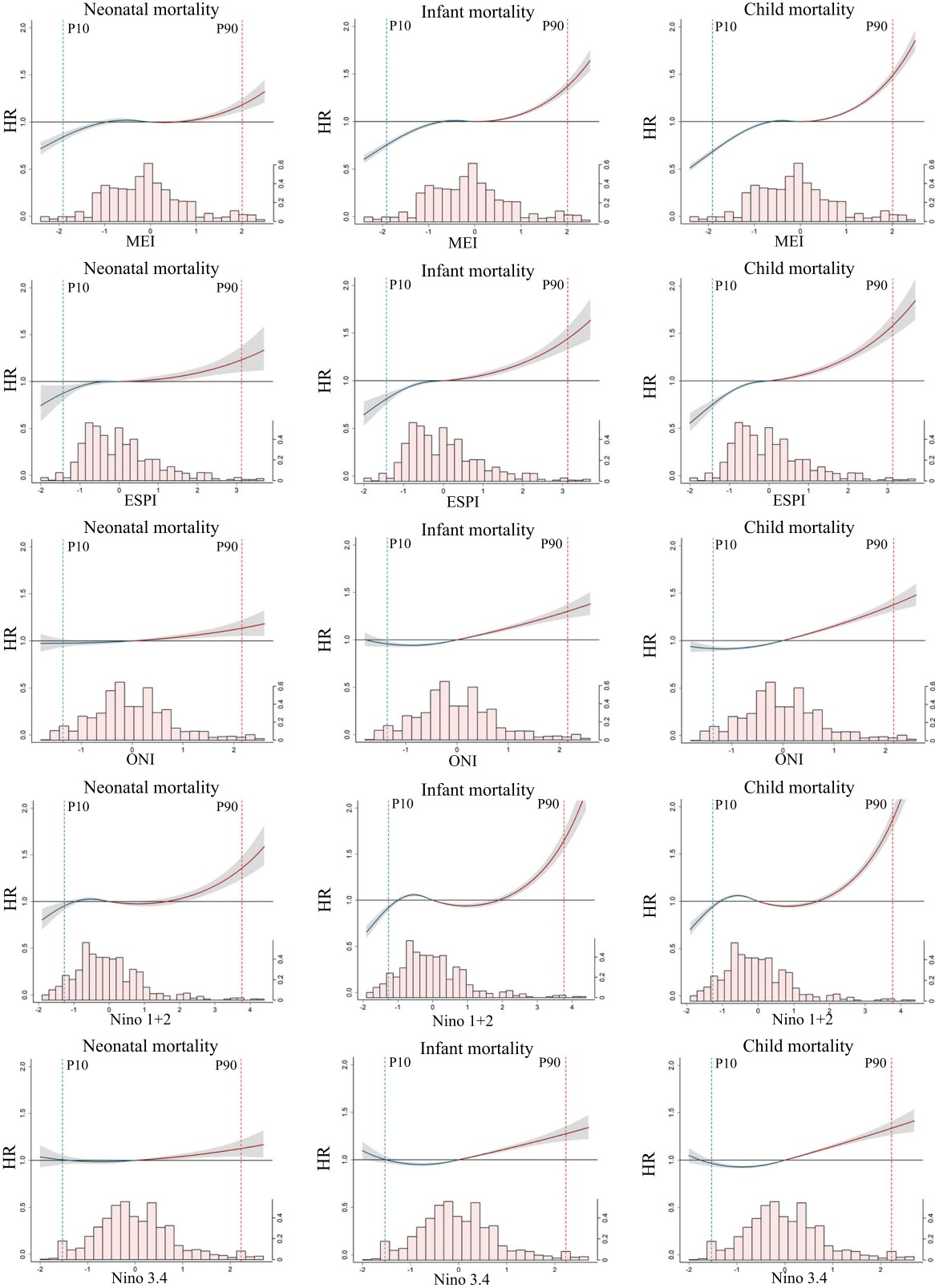

**Fig. 2 | Cumulative exposure–response associations between child survival and ENSO at 0–12 preconceptional and prenatal months of mothers.** The red and blue solid lines (with 95% confidence interval, shaded gray) indicate effect estimates of El Niño-like and La Niña-like conditions, respectively. They are the centers for the error bands. The association estimate of each ENSO measure with child survival is computed as the HR of a given percentile of ENSO relative to the reference value (set at zero). Histograms of ENSO indices are plotted at the bottom, with density measured by the second (right) vertical axis. HR hazard ratio, ENSO El Niño Southern Oscillation, MEI multivariate El Niño index, ESPI ENSO precipitation index, ONI oceanic Niño index, P90 90th percentile, P10 10th percentile.

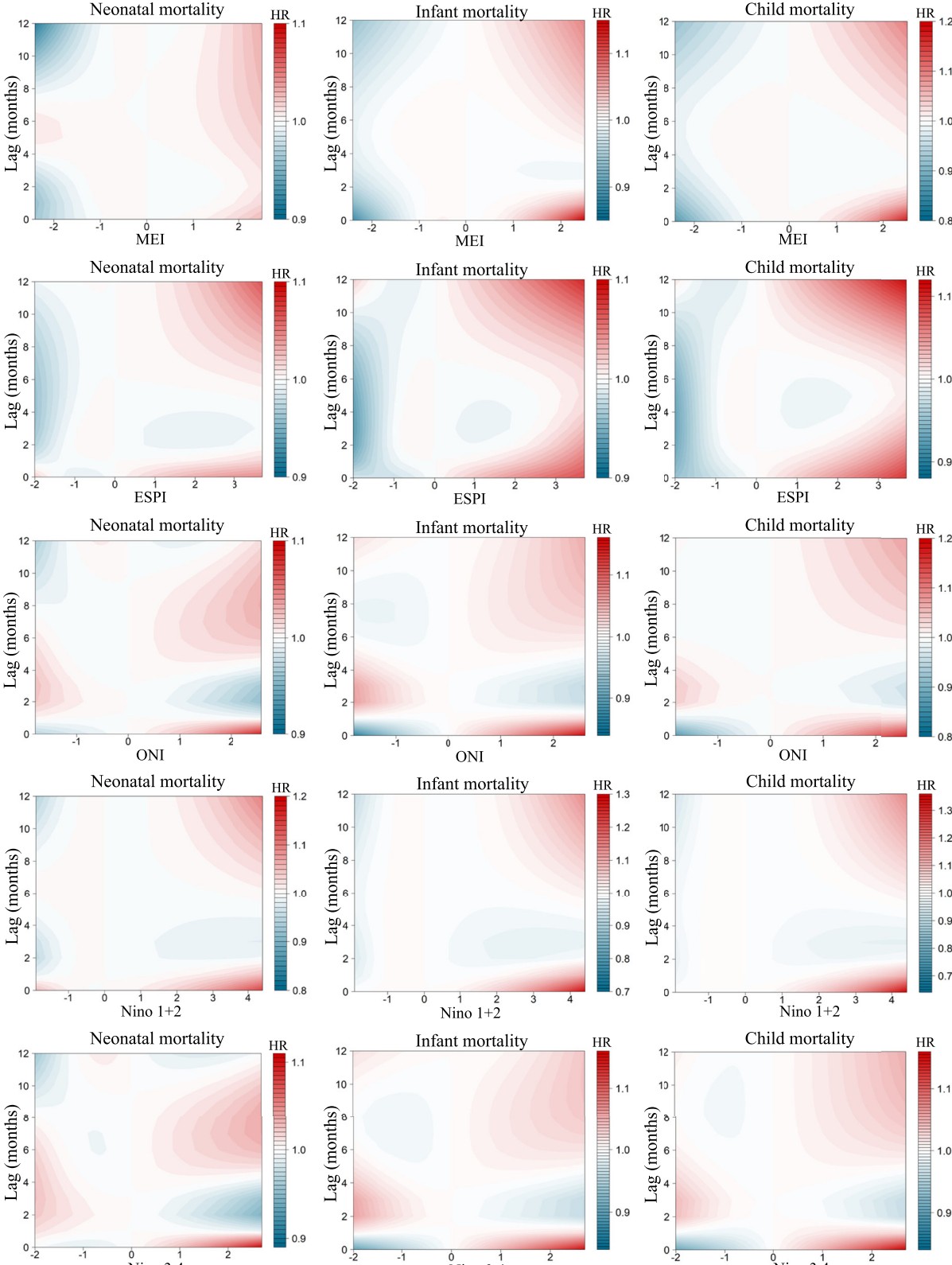

**Fig. 3 | Lag association pattern between mothers' preconceptional and prenatal ENSO (up to 12 months before delivery) and child survival from contour plots.** The *X*-axis indicates the intensity of each ENSO measure, and *Y*-axis indicates lags of 0–12 months. The color gradient represents the hazard ratio (HR). ENSO El Niño Southern Oscillation, MEI multivariate El Niño index, ESPI ENSO precipitation index, ONI oceanic Niño index.

subgroup analyses were performed to examine the associations between ENSO and mortality across tertiles of meteorological parameters at each DHS cluster location. As shown in Supplementary Table S5, the associations with ESPI and Niño 1 + 2 were greater in areas with high levels of precipitation, but no significant modification effects were found across subgroups defined by temperature levels.

Supplementary Tables S6 and S7 provide the estimated attributable fraction (AF), a measure used to quantify the proportion of

**El Niño conditions (based on 90th percentile of MEI measure)**

| Characteristics | Subgroup | Infant mortality | | | Under-five mortality | | |
|---|---|---|---|---|---|---|---|
| | | HR (95%CI) | Effect modification | | HR (95%CI) | Effect modification | |
| Sex | | | | | | | |
| | Male | 1.35 (1.27,1.42) | Ref. | | 1.47 (1.40,1.54) | Ref. | |
| | Female | 1.39 (1.31,1.48) | 0.41 | | 1.50 (1.42,1.58) | 0.54 | |
| Birth order | | | | | | | |
| | First | 1.36 (1.25,1.47) | Ref. | | 1.50 (1.39,1.61) | Ref. | |
| | Second | 1.40 (1.27,1.54) | 0.63 | | 1.53 (1.41,1.67) | 0.67 | |
| | Third or fourth | 1.29 (1.19,1.40) | 0.39 | | 1.41 (1.31,1.52) | 0.24 | |
| | ≥Fifth | 1.38 (1.28,1.49) | 0.74 | | 1.45 (1.36,1.55) | 0.55 | |
| Mother age | | | | | | | |
| | [15, 19) | 1.39 (1.28,1.52) | Ref. | | 1.54 (1.43,1.67) | Ref. | |
| | [20, 24) | 1.37 (1.27,1.48) | 0.78 | | 1.52 (1.42,1.62) | 0.75 | |
| | [25, 29) | 1.29 (1.18,1.41) | 0.21 | | 1.42 (1.31,1.53) | 0.13 | |
| | [30, 34) | 1.34 (1.20,1.50) | 0.60 | | 1.37 (1.25,1.51) | 0.06 | |
| | ≥35 | 1.42 (1.27,1.59) | 0.81 | | 1.50 (1.36,1.66) | 0.67 | |
| Urbanicity | | | | | | | |
| | Urban | 1.22 (1.12,1.34) | Ref. | | 1.35 (1.25,1.46) | Ref. | |
| | Rural | 1.41 (1.35,1.48) | 0.005 | | 1.52 (1.46,1.58) | 0.01 | |
| Child delivery place | | | | | | | |
| | Institution | 1.31 (1.23,1.40) | Ref. | | 1.43 (1.35,1.52) | Ref. | |
| | Home | 1.40 (1.32,1.47) | 0.14 | | 1.51 (1.44,1.58) | 0.20 | |
| Birth quarter | | | | | | | |
| | Q1 | 1.31 (1.19,1.43) | Ref. | | 1.38 (1.28,1.50) | Ref. | |
| | Q2 | 1.30 (1.20,1.40) | 0.91 | | 1.44 (1.34,1.54) | 0.49 | |
| | Q3 | 1.43 (1.29,1.59) | 0.20 | | 1.53 (1.39,1.68) | 0.13 | |
| | Q4 | 1.48 (1.34,1.64) | 0.07 | | 1.57 (1.43,1.72) | 0.05 | |
| Safe water | | | | | | | |
| | Yes | 1.28 (1.19,1.38) | Ref. | | 1.40 (1.31,1.50) | Ref. | |
| | No | 1.40 (1.33,1.47) | 0.06 | | 1.51 (1.45,1.58) | 0.07 | |
| Education | | | | | | | |
| | No education | 1.36 (1.28,1.44) | Ref. | | 1.46 (1.39,1.53) | Ref. | |
| | Primary | 1.45 (1.35,1.57) | 0.17 | | 1.63 (1.53,1.74) | 0.007 | |
| | Secondary or higher | 1.20 (1.09,1.32) | 0.03 | | 1.26 (1.15,1.38) | 0.006 | |
| Household wealth | | | | | | | |
| | Poorest or poorer | 1.37 (1.29,1.45) | Ref. | | 1.46 (1.39,1.54) | Ref. | |
| | Middle or richer | 1.43 (1.33,1.52) | 0.36 | | 1.57 (1.48,1.66) | 0.07 | |
| | Richest | 1.27 (1.13,1.43) | 0.27 | | 1.40 (1.26,1.56) | 0.51 | |
| Geographic zone | | | | | | | |
| South, Southeast and Western Asia | | 1.13 (1.04,1.22) | Ref. | | 1.11 (1.03,1.20) | Ref. | |
| Northern and Western Africa | | 1.23 (1.15,1.32) | 0.09 | | 1.27 (1.20,1.35) | 0.005 | |
| Central, Eastern and Southern Africa | | 1.72 (1.61,1.85) | <0.001 | | 1.86 (1.75,1.98) | <0.001 | |

HR ( 95%CI) HR ( 95%CI)

**Fig. 4 | Cumulative associations between child mortality and extreme levels of ENSO exposure at 0–12 preconceptional and prenatal months of mothers stratified by characteristics of study participants.** The sample sizes used to derive statistics are provided in Table 1. The effect estimate of each ENSO measure on child survival computed as the HR of a given percentile of ENSO relative to the reference value (set at zero) is presented as the center of error bars, and the error bars represent 95% confidence intervals of the estimates. HR hazard ratio, ENSO El Niño Southern Oscillation, MEI multivariate El Niño index. The differential association estimates between a subgroup and the reference (ref.) are tested using the two-sided two-sample z-test given by Eq. (1). The z-statistics in the last row are 7.99 (left) and 10.56 (right), and the exact p values are too close to 0 to calculate. Adjustments are not made for multiple comparisons.

health outcomes (e.g., child deaths) in the population that can be attributed to a specific exposure (e.g., a high ENSO level). The formula is provided in the "Methods" section. According to the United States National Oceanic and Atmospheric Administration (NOAA) operational definitions for El Niño and La Niña conditions, we assessed the health impacts of ENSO measures meeting or exceeding ±0.5 °C, compared to ENSO-neutral status. We found that most of the ENSO-associated mortality burden could be attributable to the contribution of El Niño-like conditions, with AF ranging from 3.7% to 20.9%. For instance, relative to the ENSO-neutral conditions based on MEI (−0.5 to +0.5 °C), the HR of infant mortality was 0.98 (95% CI, 0.96, 0.99) for La Niña (≤ −0.5 °C), 1.19 (95% CI, 1.16, 1.22) for weak El Niño (≥ +0.5 to < +1.0 °C), and 1.21 (95% CI, 1.17, 1.24) for moderate El Niño (≥ +1.0 °C). As a result, maternal preconceptional and prenatal El Niño exposures were responsible for a large portion of the child mortality burden, whereas the exposure to La Niña prior to mothers' delivery had relatively negligible AFs. Weak and moderate El Niño contributed to 15.9% (95% CI, 13.5%, 18.1%) and 17.0% (95% CI, 14.7%, 19.2%) of infant deaths, respectively. Note that, since there may also be risks in the "neutral" conditions based on Fig. 2, using this

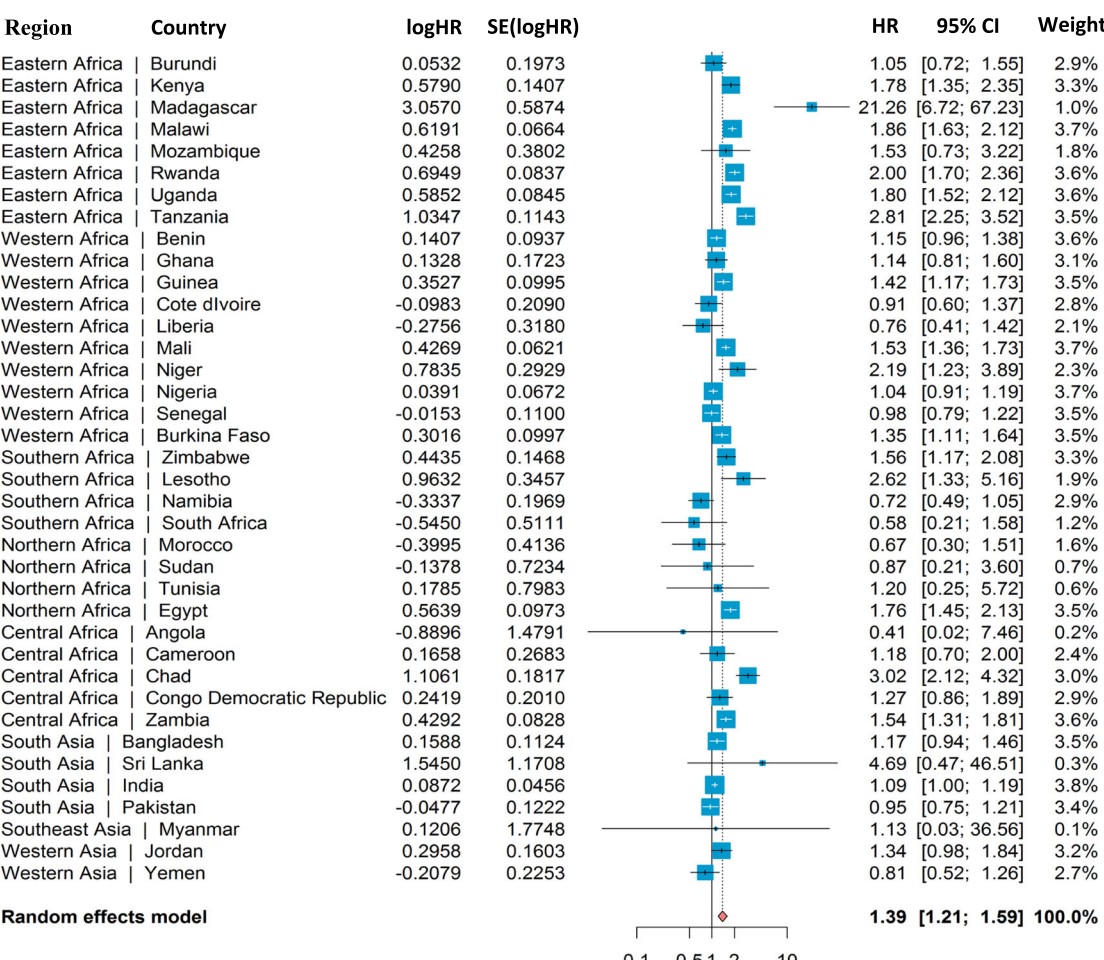

**Under-five mortality and El Niño conditions (based on 90th percentile of MEI measure)**

| Region | Country | logHR | SE(logHR) | | HR | 95% CI | Weight |
|---|---|---|---|---|---|---|---|
| Eastern Africa | Burundi | 0.0532 | 0.1973 | | 1.05 | [0.72; 1.55] | 2.9% |
| Eastern Africa | Kenya | 0.5790 | 0.1407 | | 1.78 | [1.35; 2.35] | 3.3% |
| Eastern Africa | Madagascar | 3.0570 | 0.5874 | | 21.26 | [6.72; 67.23] | 1.0% |
| Eastern Africa | Malawi | 0.6191 | 0.0664 | | 1.86 | [1.63; 2.12] | 3.7% |
| Eastern Africa | Mozambique | 0.4258 | 0.3802 | | 1.53 | [0.73; 3.22] | 1.8% |
| Eastern Africa | Rwanda | 0.6949 | 0.0837 | | 2.00 | [1.70; 2.36] | 3.6% |
| Eastern Africa | Uganda | 0.5852 | 0.0845 | | 1.80 | [1.52; 2.12] | 3.6% |
| Eastern Africa | Tanzania | 1.0347 | 0.1143 | | 2.81 | [2.25; 3.52] | 3.5% |
| Western Africa | Benin | 0.1407 | 0.0937 | | 1.15 | [0.96; 1.38] | 3.6% |
| Western Africa | Ghana | 0.1328 | 0.1723 | | 1.14 | [0.81; 1.60] | 3.1% |
| Western Africa | Guinea | 0.3527 | 0.0995 | | 1.42 | [1.17; 1.73] | 3.5% |
| Western Africa | Cote dIvoire | -0.0983 | 0.2090 | | 0.91 | [0.60; 1.37] | 2.8% |
| Western Africa | Liberia | -0.2756 | 0.3180 | | 0.76 | [0.41; 1.42] | 2.1% |
| Western Africa | Mali | 0.4269 | 0.0621 | | 1.53 | [1.36; 1.73] | 3.7% |
| Western Africa | Niger | 0.7835 | 0.2929 | | 2.19 | [1.23; 3.89] | 2.3% |
| Western Africa | Nigeria | 0.0391 | 0.0672 | | 1.04 | [0.91; 1.19] | 3.7% |
| Western Africa | Senegal | -0.0153 | 0.1100 | | 0.98 | [0.79; 1.22] | 3.5% |
| Western Africa | Burkina Faso | 0.3016 | 0.0997 | | 1.35 | [1.11; 1.64] | 3.5% |
| Southern Africa | Zimbabwe | 0.4435 | 0.1468 | | 1.56 | [1.17; 2.08] | 3.3% |
| Southern Africa | Lesotho | 0.9632 | 0.3457 | | 2.62 | [1.33; 5.16] | 1.9% |
| Southern Africa | Namibia | -0.3337 | 0.1969 | | 0.72 | [0.49; 1.05] | 2.9% |
| Southern Africa | South Africa | -0.5450 | 0.5111 | | 0.58 | [0.21; 1.58] | 1.2% |
| Northern Africa | Morocco | -0.3995 | 0.4136 | | 0.67 | [0.30; 1.51] | 1.6% |
| Northern Africa | Sudan | -0.1378 | 0.7234 | | 0.87 | [0.21; 3.60] | 0.7% |
| Northern Africa | Tunisia | 0.1785 | 0.7983 | | 1.20 | [0.25; 5.72] | 0.6% |
| Northern Africa | Egypt | 0.5639 | 0.0973 | | 1.76 | [1.45; 2.13] | 3.5% |
| Central Africa | Angola | -0.8896 | 1.4791 | | 0.41 | [0.02; 7.46] | 0.2% |
| Central Africa | Cameroon | 0.1658 | 0.2683 | | 1.18 | [0.70; 2.00] | 2.4% |
| Central Africa | Chad | 1.1061 | 0.1817 | | 3.02 | [2.12; 4.32] | 3.0% |
| Central Africa | Congo Democratic Republic | 0.2419 | 0.2010 | | 1.27 | [0.86; 1.89] | 2.9% |
| Central Africa | Zambia | 0.4292 | 0.0828 | | 1.54 | [1.31; 1.81] | 3.6% |
| South Asia | Bangladesh | 0.1588 | 0.1124 | | 1.17 | [0.94; 1.46] | 3.5% |
| South Asia | Sri Lanka | 1.5450 | 1.1708 | | 4.69 | [0.47; 46.51] | 0.3% |
| South Asia | India | 0.0872 | 0.0456 | | 1.09 | [1.00; 1.19] | 3.8% |
| South Asia | Pakistan | -0.0477 | 0.1222 | | 0.95 | [0.75; 1.21] | 3.4% |
| Southeast Asia | Myanmar | 0.1206 | 1.7748 | | 1.13 | [0.03; 36.56] | 0.1% |
| Western Asia | Jordan | 0.2958 | 0.1603 | | 1.34 | [0.98; 1.84] | 3.2% |
| Western Asia | Yemen | -0.2079 | 0.2253 | | 0.81 | [0.52; 1.26] | 2.7% |
| **Random effects model** | | | | | **1.39** | **[1.21; 1.59]** | **100.0%** |

0.1   0.5 1 2   10

Heterogeneity: $I^2 = 85\%$, $\tau^2 = 0.1245$, $p < 0.01$

**Fig. 5 | Results of cumulative associations between child mortality and mothers' exposure to El Niño conditions at 0–12 preconceptional and prenatal months from country-specific analyses and meta-analyses.** The country-specific sample sizes used to derive statistics are provided in Supplementary Table S2. The effect estimate, which is presented as the center of the error bars in the figure, is computed as the HR of 90th percentile level of MEI measure relative to the reference value (set at zero). The error bars represent the 95% confidence intervals of estimates. HR hazard ratio, ENSO El Niño Southern Oscillation, MEI multivariate El Niño index. The heterogeneity is measured by Cochran's Q, which is calculated as the weighted sum of squared differences between country-specific effects and the pooled effect across studies. The Q-statistic is 254.16; with the degree of freedom being 37, the exact $p$ value is $3.30 \times 10^{-34}$, which is way smaller than 0.01. Adjustments are not made for multiple comparisons.

reference may underestimate the AF and thus, provide more conservative estimates.

Additionally, as shown in Fig. 5, and Supplementary Figs. S7–S11, our random effects meta-analysis investigated the impacts of mothers' preconceptional and prenatal exposure to ENSO on child survival for each of our 38 LMICs, and compared country-specific estimates against the combined estimates of meta-analysis. The I-squared ($I^2$) statistic, a commonly used heterogeneity measure, was applied to describe the percentage of variability in meta-analysis; an $I^2$ value of ≤25% indicates a low level of heterogeneity, a value around 50% indicates a moderate level of heterogeneity, and a value near 75% or greater corresponds to a high level of heterogeneity[20,21]. Findings from our analyses indicated that moderate to high levels of heterogeneity were observed for the associations between mothers' exposure to selected ENSO measures (e.g., MEI) and child mortality across countries, with the values of $I^2$ statistics ranging from 53% to 85%. Countries more intensely affected by high levels of ENSO were Kenya, Rwanda, Tanzania, Uganda, Madagascar, Malawi, and Zimbabwe. Moreover, the combined estimates were similar to our primary results–for example, in Fig. 5, comparing the P90 level of MEI to the reference value at lag

0–12 months, the HR of U5M was 1.39, although with a wider 95% CI (1.21, 1.59).

Sensitivity analyses indicated the overall results to be robust. As in Supplementary Figs. S12 and S13, we found significantly increased mortality risks in association with higher levels of ENSO measures during both preconceptional (lag 9–12) and prenatal (lag 0–8) months prior to mothers' delivery. The association patterns were also generally consistent with our observations in the main models when including random effects for countries or introducing a random effect for each DHS survey cluster (Supplementary Figs. S14 and S15).

## Discussion

In this study, we used a large multi-country dataset to examine the relationship between climate anomalies originating from the equatorial Pacific Ocean and child survival. We have shown here that maternal exposure to ENSO during and prior to delivery, particularly El Niño-like climate changes, is significantly associated with heightened risks of all-cause child mortality (neonatal, infant, and all under-five deaths). Further, El Niño-like conditions had higher contributions to mortality burden and were responsible for 3.7%–20.9%

of mortality in children under 5 years. Subgroup analyses indicated that poor socioeconomic status and quality of health services, including lower maternal education levels and unsafe sources of drinking water, modified the linkages between El Niño-like conditions and child mortality. More specifically, the effect estimates were greater in participants residing in rural areas and from Central, Eastern, and Southern Africa. Our findings support the possibility that large-scale climate anomalies driven by ENSO can be a determinant of child survival, which might have critical implications for reducing U5M at a global scale.

Although the associations between ENSO exposure and child survival had not previously been systematically assessed, a cross-sectional study indicated that occurrences of under-two child mortality peaked during El Niño events[22]. Nevertheless, whether these phenomena could be explained by variations in the intensity of ENSO metrics warrant further investigations, and well-documented exposure–response relationships remain largely sparse. To our knowledge, prior quantification studies of the ENSO effects on mortality have generally focused on the adult population[15,16]. Results from the annual aggregate health data of older adults living in North America indicated that significant increases in risks of annual morbidity and mortality rates were observed in association with even weak to moderate El Niño events among participants from the western United States, although no effects were observed in those residing in the southern United States and Canada[15]. One possible explanation of the findings is that the annual aggregation of death data irrespective of known ENSO-associated dynamic climate anomalies might have limited the ability to capture the ENSO's teleconnection impacts. A recent investigation at the individual level that linked ENSO with a variety of health data in Chinese elderly showed that both El Niño and La Niña-like conditions were closely correlated with mortality risks, and U-shaped relationships were found in their analysis based on recent advances in statistical methods such as distributed lag non-linear models (DLNM)[16]. They also found that some individuals might be more prone to the mortality effects attributable to El Niño-like conditions. Using DLNM allowed them to quantify a comprehensive time-course picture regarding the exposure–response relationship[23]. Herein, our analytical approach introduced similar models to associate U5M with ENSO in a retrospective cohort study of young children with adjustments for crucial confounding variables. We observed approximately J-shaped associations between maternal ENSO exposure during and prior to delivery and child mortality, with stronger impacts attributable to more El Niño-like conditions. The findings were also confirmed by the regressions of ENSO measures as categorical variables. Moderate El Niño showed the strongest associations with child mortality, followed by weak El Niño, whereas La Niña exhibited null or negative associations when compared with the neutral conditions. Collectively, the current study provides evidence that extreme levels of ENSO can be a trigger of worsening child survival.

We also evaluated the lag association pattern with child survival to understand sensitive windows of maternal exposure. The hypothesis of Developmental Origins of Health and Disease states that unfavorable in-utero life exposures can exert maladaptations that influence the whole life course, including morbidity and mortality later in life[24]. Our findings emphasized that this influence could arise early in life and even contribute to infant and neonatal deaths. We found increased risks of child mortality primarily in mothers exposed to El Niño-like climate changes at various consecutive lagged months prior to birth giving, especially in the periods of the 0th–1st and 6th–12th lagged months. To date, the intensity of climate anomalies driven by El Niño could be predicted at least 9 months in advance, suggesting that combining the sensitive windows of exposure with early-warning systems might help address potential climate change crises for child health through developing appropriate preventive measures and

adaptive strategies. Moreover, our results of AFs indicated that El Niño could be partially responsible for the heightened risk of all-cause child mortality. We acknowledge that the magnitude of association estimates derived from El Niño observed in the present analysis might seem relatively modest; however, its detrimental effects could be relevant to hundreds of millions of reproductive-age women and young children because climate-related environmental stresses, in numerous regions around the world, have been shown to be teleconnected with frequent occurrence of El Niño episodes[8].

In this pooled analysis for multiple LMICs, we also discerned several potential effect modifiers regarding the socioeconomic status and quality of health services. It appears that the child mortality risks attributable to El Niño-like conditions were smaller in mothers residing in urban areas and those with high education levels. Individual-level education has served as the most commonly used indicator reflecting socioeconomic status in epidemiologic studies[25]. Therefore, these differences within subgroups might be because rural/lower income individuals have limited access to using appropriate facilities against El Niño events, possibly due to economic burden. We further revealed that the mortality effects posed by El Niño were greater in those who had poor sources of drinking water, which implied synergistic effects between unhealthy hygiene practices and extreme ENSO for child survival. Compared with the South, Southeastern, and Western Asia, participants from Africa had a greater mortality burden. One possible explanation for this finding is that variations in ENSO could disproportionately affect regional weather[26]. Taken together, we extended existing evidence by empirically showing that several susceptibility factors, such as poverty and unsafe drinking water can exacerbate the deleterious effects of climate anomalies.

Another crucial result of the present study is the significant differences in the associations between preconceptional and prenatal exposure to ENSO and child survival across our study areas. For instance, while some regions such as central, eastern, and southern parts of Africa reported the largest risk estimates (i.e., HR) above 2.4, relatively smaller estimates were observed in south, southeast, and western parts of Asia (with HR below 1.2) and Northern and Western Africa (with HR below 1.9). Several factors, such as population characteristics, may serve as possible modifiers of the observed associations; but these might not fully explain the differential associations between regions. More recently, given the consequences of El Niño between July and September in 2023, the WHO has identified several high-risk countries in Eastern Africa and recommended some close-monitoring countries in Southern Africa[11]. The countries potentially enduring greater impacts of El Niño were Kenya, Rwanda, Tanzania, Uganda, Madagascar, Malawi, and Zimbabwe[11], which was consistent with our country-specific findings in the meta-analysis. It has been pointed out that, wet and dry conditions regulated by ENSO teleconnections may be partially responsible for the differential associations across the study areas[11]. In Eastern Africa, El Niño is typically associated with higher-than-normal rainfall and increased risk of flooding, thereby contributing to river overflows downstream, population displacement, crop and livestock losses[27]. Indeed, positive correlations between selected ENSO measures (e.g., MEI) and local meteorological parameters (e.g., precipitation) were found in most areas of Eastern Africa in the present analysis. Further, water contamination caused by heavy rainfall and flooding could exacerbate and prolong the outbreaks of climate-sensitive diseases[6]. In contrast, El Niño events are associated with drier-than-normal conditions in Southern Africa, and significant food security impacts due to droughts-related reductions in food production may occur in this region[28,29]. All of these El Niño climatic impacts are likely key substrates for worsening maternal and fetal health, eventually leading to increasing the risk of child death.

Various mechanisms have been implicated in the pathogenesis of child mortality risks associated with exposure to climate-related

environmental stressors[11,12]. El Niño conditions may enhance the probability of droughts, floods, fires, and heat waves, all of which are deleterious to child survival[30]. The WHO has proposed that water-borne diseases diarrhea, cholera, and malnutrition are among the most climate change-sensitive health responses[19]. Drier-than-normal conditions, flooding, and above-normal rainfall are capable of restricting access to safe water, and flooded sanitation infrastructure can result in water-borne diseases[12]. It has been shown that El Niño is linked to outbreaks of cholera and other diarrheal diseases, possibility through flooding-related contaminated water[11]. Likewise, the relationships between El Niño events and vector-borne diseases (e.g., malaria) have been well corroborated in Africa and parts of Asia[30]. El Niño is also expected to change in dynamics of weather patterns, thereby heightening risks of gastrointestinal infections posed by bacterial pathogens in water[17]. Life-threatening manifestations of diarrhea during pregnancy include dehydration and undernutrition, which are due to inhibiting nutrient absorption in gastrointestinal tracts, contributing to weakened immune systems in both mothers and fetuses[31,32]. El Niño-induced malnutrition might be a consequence of El Niño's impacts on both food security and elevation of diarrheal diseases[33]. Vulnerable individuals such as women and their infants in marginalized communities may be at higher risks[34]. A study reported that children born during and after the 1997–1998 El Niño event experienced negative impacts on their long-term nutritional status, characterized by shorter height and less lean mass[35]. Many studies have also shown that maternal undernutrition is associated with low birthweight of children, and underweight is in turn an important predictor for child mortality[36]. This evidence suggests a potential link between El Niño and undernutrition among both mothers and infants, which may lead to child underweight and infections, increasing child mortality in the longer term. Additionally, both hot and dry conditions can trigger heat waves, wildfire smoke, poor air quality, consequently causing exhaustion cardiorespiratory diseases and heat stroke as well as exacerbating underlying illnesses resulting in premature death[11,37]. In the context of climate change, global temperature levels are boosted by extreme levels of ENSO, and Southern Africa has recorded El Niño event-associated extreme heat in 2015–2016[11]. Indeed, pregnant women are typically at high risk of direct heat exposure due to hormonal changes during pregnancy[19]. Also, abnormal temperature exposure during in-utero development has been found to be associated with elevated risks of low birthweight, preterm birth, and child death[38]. We have previously shown that the effects of El Niño-like conditions on mortality among elderly individuals from areas with higher ambient temperature[16]. In this study, the associations of child mortality with the P90 of several ENSO measures were slightly stronger in participants residing in areas with high levels of precipitation, but no significant differences were observed across subgroups defined by the levels of temperature. These collective findings suggested that the detrimental effects attributable to El Niño-like conditions might be affected by weather patterns. Future investigations on the exact mechanisms responsible for the association between El Niño and worsening child survival are urgently warranted.

Our study has several strengths. To our knowledge, the environmental determinants of child mortality from the perspective of ENSO-driven climate anomalies remain to be characterized. Herein, we constructed a continentally large dataset of children under 5 years old in multiple LMICs based on the well-established DHS. Using Cox regressions coupled with DLNM allowed us to characterize the impact of preconceptional and prenatal exposure to ENSO on U5M in children and identify susceptible exposure windows. Nevertheless, some limitations in our study should be noted. First, although a wealth of covariates, including demographic and socioeconomic characteristics, were considered in the main models based on the existing literature and biological mechanisms, we are not able to rule out the possibility of residual confounding posed by potentially unmeasured covariates

in this observational investigation. Second, given that the information on some covariates was obtained from children's mothers at the end of their interview, potential recall bias or misclassification might exist. Third, we did not conduct cause-specific analyses because data on the cause of death were not available in the surveys. Also, further investigations are warranted to assess the impact of ENSO exposure on the full spectrum of health outcomes in children, thereby fully capturing the detrimental health effects posed by ENSO. Fourth, although this retrospective cohort study was performed by pooling analysis of more than one million individuals, which could yield sufficient statistical power on assessing the overall relationship between ENSO exposure and child survival, given that we generated a good precision (narrow CI) around the point estimates, it must be acknowledged that the study sample may not be representative of some conflict-affected areas in countries experiencing wars or insecurity, or countries that opt out of participating in the DHS[39]. The magnitude of this under-representativeness of conflict-affected areas may vary from one country to another. For instance, in the Democratic Republic of the Congo (or Congo Democratic Republic), only 4 of the 540 sampled clusters were not accessible due to insecurity in the 2012/2014 DHS surveys (that is, <1% all sampled clusters were affected). Therefore, we speculated that conflict conditions were not likely to have substantial influences on the magnitude and direction of the associations observed in this study. Noteworthy was that this speculation was derived based only on the example of the Democratic Republic of Congo, given that we were unable to obtain comparable data for other countries. Fifth, we should also limit our interpretations of the results to the regions shown in Fig. 1, and how the meteorologic parameters serve as both mediators and moderators requires further investigation. Notwithstanding, climate-related stresses can exacerbate health inequalities around the world, and participants residing in LMICs would bear a larger burden of ENSO exposure due to systemic economic injustices, such as poverty and poor environmental conditions and hygiene practices[40]. We have recognized that this supposition should be tested in future investigations in more LMICs with ENSO tele-connected impacts.

In summary, our multi-country analysis presented exposure–response relationships and association patterns of pre-conceptional and prenatal exposure to ENSO and child survival, with the important implication that maternal exposure to extreme levels of ENSO may be a crucial environmental risk factor for U5M. The magnitude of such effects was greater at the 0th–1st and 6th–12th lagged months prior to child delivery and worsened by poor socioeconomic status and quality of public health infrastructures. These findings highlight the urgent need for better early-warning systems for El Niño events and aggressive community-based interventions (e.g., improvements in drinking water, mitigating crop loss) to prevent ENSO-related climate anomalies and change-attributed mortality in young children, particularly those from LMICs.

## Methods

### Study design and study population
We constructed a multi-country population-based retrospective cohort study of children under 5 years of age using 160 repeated cross-sectional surveys from 38 countries fielded between 1985 and 2018. The DHS is an ongoing collaboration program between the United States Agency for International Development and country-specific institutes to perform nationally representative household surveys in LMICs[41]. The general goal was to expand on the World Fertility Survey to monitor a variety of population health measures, with a focus on demographic, fertility, and family planning of reproductive-aged women and their young children. DHS data are available free of charge from the program website (https://dhsprogram.com/) after researchers register a user account online, provide a brief description of the proposed projects, and receive the approval from the DHS

Program. Detailed information on the design of the DHS has been previously described and can also be found on the program website[41]. Briefly, DHS has been administered in multiple LMICs since the 1980s, and nearly all DHS samples were representative at the national level. Reproductive-age women within per primary sampling unit (namely, a DHS cluster location, with ~20−30 households) were randomly selected using a two-stage cluster sampling design, which employed enrollment processes stratified by geographical region and urban/rural location. All DHS respondents answered a suite of general questions about household- and individual-level covariates and each child's records at birth or the time of the interview, no matter if the child was still alive.

In this study, we included all the publicly available standard surveys facilitated by IPUMS-DHS before October 15, 2022 after excluding countries that did not record calendar dates for interviews to avoid comparability issues. IPUMS is an online data dissemination tool (https://www.idhsdata.org/idhs/) that has harmonized various variable names and codes across many DHS samples, making it easier to compile the relevant information on different types of data and perform comparative research comprehensively. A total of 38 LMICs including 31 African and 7 Asian countries were selected, covering nearly 3.1 billion people (38.9% of the total population) across the globe in 2021. To enhance international comparability of reproductive-age women, we restricted the analytic sample to children whose mothers were 15−49 years old when giving birth, which excluded about 0.4% of the original sample. We further dropped any child born more than 5 years preceding or after the first day of the mother's interview, which accounted for <0.1% of the original sample. As shown in Supplementary Tables S1 and S2, the enrolled countries could be further categorized into eight geographic regions based on the definition of the United Nations (https://unstats.un.org/unsd/methodology/m49/), including Central Africa (Angola, Cameroon, Chad, Congo Democratic Republic, and Zambia), Eastern Africa (Burundi, Kenya, Madagascar, Malawi, Mozambique, Rwanda, Uganda, and Tanzania), Northern Africa (Morocco, Sudan, Tunisia, and Egypt), Southern Africa (Lesotho, Namibia, South Africa, and Zimbabwe), Western Africa (Benin, Ghana, Guinea, Cote d'Ivoire, Liberia, Mali, Niger, Nigeria, Senegal, and Burkina Faso), Western Asia (Jordan, Yemen), South Asia (Bangladesh, Sri Lanka, India, Pakistan), and Southeast Asia (Myanmar). Given that observations in DHS from Sudan, Tunisia, Yemen, and Sri Lanka were missing some important variables (e.g., household wealth), these countries were excluded from the primary analysis but remained in our meta-analysis. Additionally, because the proportion of missing data for all covariates of potential interests in the remained observations was under 10% (see Supplementary Table S3), we did not implement any imputation approach. Therefore, after excluding individuals with missing data, our primary analytic sample consisted of 1499,727 de-identified children under 5 years of age born between 1981 and 2018 across 34 countries based on all available data. Since the study concerns non-human subject research, consent from participants was waived and IRB approval was not required for this study. Our research complies with all relevant ethical regulations.

## Outcome measurement
Data on child survival status were retrospectively collected by interview surveys, including the date of birth and the age at death recorded by month. A recent report from WHO suggests that U5M rate may be a useful indicator for capturing the overall health threats posed by ENSO-related climate anomalies[11]. We, therefore, considered three tiers of all-cause mortality for children as our outcome variable following the existing literature[42], including all under-five (death at <60 months of age), infant (death at <12 months of age), and neonatal (death at <1 month of age) mortality. Follow-up periods were defined as child age at interview for kids who survived all these months of life after birth, and child age at death for 103,557

deceased kids, resulting in 39,987,500 person-months in our primary analytic dataset.

## Covariates
Information on child-level, mother-level, and household-level covariates that may confound the impacts of ENSO on mortality was collected by self-report through face-to-face interviews with each DHS respondent. The potential covariates included (1) child-level characteristics: sex of the child (male versus female) and birth order (first, second, third, or fourth, versus ≥fifth born); (2) mother-level characteristics: maternal age at childbirth, the completed educational level of mother, child delivery location, maternal marital status (in marriage versus others, such as widowed, divorced, separated, and never married); (3) household-level characteristics: place of residence (rural versus urban), access to toilets in the households (having toilet facility versus no facility), access to safe water, and household wealth. Education status of a mother was classified into three response categories: no education, primary, and secondary/higher level. Child delivery location was assessed through the question, "where was the child delivered?", and responses were grouped into two categories: home versus institution (including public sector facilities, private medical sector facilities, non-governmental organizations, religious/mission facilities). Access to safe water was measured by the type of the main sources of drinking water of households, and the responses were classified into two categories: yes (i.e., water sources from piped tool and purchased from supplier) versus no (i.e., well surface water and rainwater). Household wealth as a quantile index was generated in collaboration with the World Bank, and the wealth index was computed based on household data available on asset holdings using the approaches of Filmer and Prittchett[41]. The wealth index levels were then grouped into three categories for further analysis, namely, bottom 40%, middle 40%, versus top 20%.

## ENSO and weather data
ENSO is a large, complex, and dynamic system, involving different aspects of the ocean and the atmosphere over the tropical Pacific. Using several different indexes derived from various measurements at sea level, such as sea surface temperature and pressure, instead of a single index, can be informative and beneficial in measuring the overall ENSO state. We collected monthly levels of several ENSO indices, including MEI, ESPI, ONI, Niño 1 + 2 abnormal index, and Niño 3.4 abnormal index, from the NOAA. For instance, MEI is an indicator reflecting variation in the intensity of ENSO, which incorporates several oceanic and atmospheric parameters, such as air pressure and sea surface temperature, over the equatorial Pacific Ocean into a comprehensive index, with measurable values ranging from −3 to +3. Lower values of ENSO indices typically indicate more La Niña-like (cold) conditions, while higher values represent more El Niño-like (warm) conditions[43]. More information on the time series of ENSO measures can be found from the NOAA website (https://psl.noaa.gov/data/climateindices) or Supplementary Table S8.

Emerging evidence suggests that ENSO-driven climate anomalies could adversely affect health through its tele-connected impacts on weather conditions (e.g., changes in regional temperature)[16]. Thus, monthly meteorologic parameters (temperature and precipitation) within a 10-km circular buffer around each DHS cluster location were directly obtained from IPUMS-DHS.

## Statistical analysis
We calculated descriptive statistics (mean [standard deviation, SD] for continuous variables and relative frequency [percentage] for categorical variables) to summarize the characteristics of study participants and outcomes. Spearman correlations were calculated across measurements of ENSO indicators and local levels of meteorologic parameters.

Associations between mothers' preconceptional and prenatal exposure to ENSO and child mortality were evaluated by Cox proportional hazard models, with survival time calculated as the period from a child's birth to the date of death or the first day of the mother's interview. In the primary analysis, we focused on the analytic sample of 1499,727 children with the data of all covariates available. Given that growing evidence has shown that relationships between mortality risks and ENSO could be non-linear, the Cox models incorporated with DLNM were therefore constructed, providing a flexible framework to characterize delayed associations[16,23]. Consistent with previous studies, our Cox models included the cross-basis function of monthly ENSO measures built by the DLNM, which were specified by B-spline using the "bs" function. Regarding the space of lags in the cross-basis function, three knots were placed at equally-spaced log scales using the "lognots" function[16,44,45]. Because certain covariates might have unequal influences on the outcome measures, we used a Least Absolute Shrinkage and Selection Operator (LASSO) regression method to select the above-mentioned covariates for each metric of mortality separately. This approach has been widely applied to account for the potential multicollinearity between covariates and to facilitate variable selections[46]. Covariates with three or more categories were first transformed into dummy variables before the LASSO regression. A generalized linear model was fitted through penalized maximum likelihood, and the minimum mean error parameter lambda ($\lambda$) was generated from cross-validation. Then, $\lambda$ was re-introduced to compute the optimal coefficients of each covariate. The fitting plot of $\lambda$ and screening variables for outcome measures were presented in Supplementary Figs. S16–S18. Apart from covariates determined by LASSO, all Cox models additionally adjusted for the country fixed effects to explain potential differences across the countries (with Angola chosen as the reference category) similar to previous studies[47,48]. Covariates regarding child-level, mother-level, and household-level factors included in fully adjusted Cox models were collected on the first day of the survey for each child. Given that previous research emphasized adverse effects of maternal and in-utero exposures on early childhood health and episodes of El Niño and La Niña typically last 9–12 months[11,24], we focused on examining the overall cumulative associations with ENSO exposure levels up to 12 months prior to the birth date of each child (covering mothers' preconceptional and prenatal periods). Further, the lag association patterns, depicted by contour plots, were performed to graphically discern potentially sensitive windows of exposure[49].

Subgroup analyses were conducted to examine the effect modification of demographic and disease susceptibility factors, including child sex, child birth order, child delivery location, maternal age group (15–19, 20–24, 25–29, 30–34, and ≥35 years old), maternal educational level, household wealth group, safe water access, residential address, birth quarter (Q1, January–March; Q2, April–June; Q3, July–September; Q4, October–December) and geographic zone (South, Southeast and Western Asia; Northern and Western Africa; and Central, Eastern, and Southern Africa). Also, we assessed whether the associations with ENSO exposure might differ in participants exposed to different levels of weather conditions. Thus, we implemented subgroup analyses across tertiles of annual average ambient temperature and precipitation during the investigation period, based on available data collected around each DHS cluster location (1068,152 participants with temperature measurements and 1108,695 participants with precipitation measurements). In addition, subgroup analyses were conducted by the type of fuel for cooking in the households based on data collected from 1284,931 participants. The type of cooking fuels may serve as a proxy for indoor air pollution exposure and was classified into two categories according to the existing literature: solid fuel (e.g., use of wood, coal, grass, straw, charcoal, shrub, dung, or agricultural crop residues) versus clean fuel (e.g., use of electricity, natural gas, liquefied petroleum gas, biogas, and solar)[50]. The differential association

estimates between two subgroups were tested using two-sample z-tests, and the formula was as follows:

$$\left(\hat{\beta}_1 - \hat{\beta}_2\right) \pm 1.96\sqrt{\left(SE_1\right)^2 + \left(SE_2\right)^2} \tag{1}$$

where $\hat{\beta}_1$ and $\hat{\beta}_2$ are the association estimates for the two subgroups, $SE_1$ and $SE_2$ are their respective standard errors[51]. To further estimate the burden of child death attributable to extreme ENSO conditions, we repeated the Cox models by regressing child mortality on categorical variables for ENSO measures in the 0–12 lagged months before birth. As previously defined by NOAA and previous studies, the levels of ENSO measures ≤ −0.5, ≥ +0.5 to < +1.0, and ≥ +1.0 were classified as La Niña, weak El Niño, and moderate El Niño, respectively[15,52]. The ENSO exposure is considered neutral when the measurable values range from −0.5 to 0.5. As deaths are usually considered rare events, HR and relative risk (RR) were assumed to be a reasonable approximation of each other in this study with caution[53–55]. Then, regression coefficients corresponding to ENSO measures were used to calculate the AF of deaths and the number of deaths attributable to extreme levels of ENSO, using commonly described approaches as follows[56]:

$$AF = \frac{RR - 1}{RR} \times 100 \tag{2}$$

where RR is approximated by the HR of La Niña, weak El Niño, or moderate El Niño conditions considering the neutral condition as the reference. We calculated the number of deaths attributable to exposure to La Niña, weak El Niño, or moderate El Niño by multiplying the AF with the number of deaths under these conditions.

A meta-analysis was further performed to not only investigate country-specific patterns but also test the robustness of our main results by a two-stage analytic strategy, an approach that may account for the hierarchical structure of the DHS data within countries. In the first stage, we estimated country-specific associations between maternal ENSO exposure and child mortality for each country using the same fully adjusted covariates as our main Cox models coupled with DLNM. Although observations from Sudan, Tunisia, Yemen, and Sri Lanka lacked the data for the household wealth index, these countries were also included in this analysis by not controlling for this covariate. In the second stage, we pooled the estimates of the country-specific associations using random effects models in meta-analyses[57], and the combined effect estimates represent the overall average ENSO-mortality associations across our 38 countries. The level of heterogeneity in country-specific estimates was measured by $I^2$ statistics. A set of sensitivity analyses were performed to test the robustness of our findings. We first separated the analyses for assessing the impact of ENSO exposure during the mothers' preconceptional and prenatal periods. Given the lack of exact information on maternal gestational age in the DHS, we assumed that each childbirth was carried to a full 9-month term as previously estimated[42,48]. We, therefore, assigned the ENSO exposure levels during the 9th–12th lagged months prior to birth as the "preconceptional" exposure periods and the 0th–8th lagged months prior to birth as "prenatal" exposure periods. Second, Cox proportional hazard models were repeatedly conducted to analyze the association with ENSO exposure by including random effects for different countries[58]. Third, the random effect for each survey cluster was also considered to adjust for cross-cluster differences in the DHS[59]. For the second and third sensitivity analyses, Cox frailty models were applied with random intercepts to account for clustering effects[60].

The results for ENSO exposure derived from our Cox models coupled with DLNM were presented as the HR with 95% CI by calculating the health impact of a given ENSO value relative to the

reference value because non-linear relationships with ENSO exposure were observed[16]. The reference value for each ENSO indicator was set at zero, which was typically close to the median value during the study period[61]. According to the interval of exposure–response curves, we calculated the effect of La Niña-like conditions in the 0–12 lagged months prior to birth on child mortality by comparing the P10 of exposure level with the reference value (zero), and the effect of El Niño-like conditions was calculated by comparing the P90 of exposure level with the reference value (zero). For results from the regression of mortality on categorical ENSO exposure, the HR and 95% CI for extreme levels of ENSO exposure group, compared with the neutral exposure group, were reported. All data analyses were conducted between October 2022 and August 2023 and analyzed using R, version 4.3.3 (R Project for statistical computing). The following packages were primarily used: package "survival" was installed to characterize associations with outcome measures; package "glmnet" was loaded to select variables by LASSO; package "dlnm" was applied to specify the cross-basis function for ENSO exposure and to predict and graphically plot the results of the fitted regression models[62].

### Reporting summary
Further information on research design is available in the Nature Portfolio Reporting Summary linked to this article.

## Data availability
The DHS data used in this study were de-identified and freely available from the program website (https://dhsprogram.com/), but third-party restrictions apply to the availability of the data. The data were used under license for this study with restrictions that do not allow for the data to be redistributed or made publicly available. To gain access to the health and meteorologic data, researchers should register a user account online, provide a brief description of the proposed projects, and receive the approval from the DHS Program at the IPUMS-DHS website (https://www.idhsdata.org/idhs/). This process can be done within a day without data-sharing agreements. The data on the monthly intensity of all the ENSO measures used in this study can be downloaded from the NOAA website at https://psl.noaa.gov/data/climateindices. The GIS boundary files used to draw the maps presented in Fig. 1A–C are publicly available from the IPUMS-International website (https://international.ipums.org/international/gis.shtml). Data supporting the findings of this study are all available in the article, in the supplementary files, and from the corresponding author upon request. Source data are provided with this paper.

## Code availability
The primary R codes can be found at https://doi.org/10.5281/zenodo.11408167.

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

## Acknowledgements

C.C.Z. acknowledges support from the Peking University School of Economics Research Seed Grant and the Fundamental Research Funds for the Central Universities, Peking University (2023 ID: 7100604271; 2024 ID: 7100604568). H.X. acknowledges support from the Beijing Natural Science Foundation (7234400) and China Postdoctoral Science Foundation (2021M690249). C.C.Z. also acknowledges support from the National Social Science Fund of China (22VMG017) and the National Natural Science Foundation of China (71973008).

## Author contributions

H.X. and C.C.Z. performed data analysis and wrote the manuscript text; H.X., C.C.Z., and X.H. conducted exposure assessment; H.X., C.C.Z., V.M.O., E.B.M., Q.Z., and W.H. interpreted relevance of study findings. All authors read and approved the final manuscript.

## Competing interests

The authors declare no competing interests.
