## [Peer Review File · Nature Communications]

Maternal Preconceptional and Prenatal Exposure to El Niño Southern Oscillation Levels and Child Mortality: A Multi-country StudyREVIEWER COMMENTS

Reviewer #1 (Remarks to the Author):

The purpose of this paper is to examine the relationship between the meteorological phenomenon known as El Niño Southern Oscillation (ENSO), and childhood mortality in a selected set of 38 low and middle income countries (so-called LMICs). ENSO is measured using one of five indicators, abbreviated MEI, ESPI, ONI, Niño 1+2 and Niño 3.4. The outcome variable is one of total under-5 mortality (U5M), infant mortality (meaning mortality under one year of age) and neo-natal mortality (under one month of age), the latter two categories accounting for respectively 77% and 43% of the total U5M. Regression models were fitted to show the relationship between ENSO and mortality using non-linear splines and possibly some additional covariates (the last point is never made completely clear). In addition, the authors performed numerous subgroup analyses. The overall conclusion is that there is a statistically significant positive relationship between U5M and high levels of El Niño, typically characterized by the difference between the 90th percentile of El Niño and the zero value of the respective index. The authors' conclusion from this is the need for better early warning systems and mitigation measures during future El Niño events.

I think this paper has merit in collecting and analyzing a large dataset from the Demographic and Health Surveys (DHS) Program and relating it to numerous demographic and climate indicators. I had not previously heard of this dataset, though I presume that the present authors are not the first to have analyzed it to look for mortality patterns; that fact alone makes this seem to me a useful exercise. However, have two major criticisms of what they do, and numerous more minor queries.

My major queries:

1. Why do they assume these mortality patterns are due specifically to El Niño and not just to the most familiar meteorological variables, temperature and precipitation? I am not very familiar with the literature on childhood mortality but for adult mortality (e.g. studies based on the US Medicare database, which is mostly adults over 65) there is a huge literature on climate-related mortality patterns, that generally supports the present authors' claim of a J-shaped curve (a non-linear relationship that curves sharply upwards at the high-temperature end of the curve but not at the low-temperature end). Without taking account of this very obvious confounder, it seems to me that claims of a causal effect are misguided.

2. I have really serious concerns about the way these authors present their statistical analysis, which go well beyond answering a few questions about fine-level details of the analysis. Throughout the paper, they are vague about what they are doing; they don't provide any of their computer code, nor do they give enough analytic detail for the reader to have any hope of reproducing their analysis. I already referred to the authors' vagueness about covariates; specifically, on page 33 they say "the following covariates were considered..." but what does "considered" mean? Were they included or not? If some were and some weren't, how did they decide which ones to include? There are standard statistical methods for deciding which covariates to include in a regression equation, from the very traditional (such as stepwise regression), through more modern methods such as ridge regression or lasso, up to machine learning techniques such as random forests or boosting - I am not necessarily implying that they should have used machine learning methods but they do need to say what they actually did. Other necessary details of the statistical analysis are only very loosely described, e.g. "quadratic B-splines with three degrees of freedom" (what exactly does this mean? - either give a formula or a reference) or "corresponding cross-basis function with three knots at equally-space log scales" (same comment). I'm also concerned about the lack of any discussion of country by country effect (they do some subgroup analyses restricting attention to "geographic zones", but they don't consider country as a possible covariate in their main analyses). This is of concern because one would expect mortality rates to differ substantially by country, and this is a possible confounder in an analysis like this.

Other queries that are less important but I would still like to see answered:

3. Why did the authors consider five different measures of El Niño and what additional information do they think they gained from doing so (as compared with the alternative strategy of just using one measure throughout the analysis)? I am aware that different meteorology research groups have different preferences on this point, and that sometimes they may prefer one measure over another based on which region of the earth they are studying, but it's not clear to me why you would want to repeat the analysis five times over with different measures. Moreover, extended Figure S1 seems to show that they all go up and down more or less together, so why make a distinction?

4. Another concern I have is the use they make of different lags in the ENSO values. I am well aware of and generally support the need to consider lags in any analysis of this nature, because in general there are lags of between a few months up to at least a year between the ENSO signal and its effects, but one has to be very careful to handle the lags in a way that does not raise suspicions of cherry-picking your results. Despite the authors' claims, I was unable to perceive a clear pattern in the various panels of Figure 3. I believe they could have taken a more systematic approach to the selection of suitable lags that would avoid the concern about selecting lags to fit their hypothesis.

5. Just as a textual point, the authors need to be more careful about defining their terminology as they go. For example, MEI and ONI are first mentioned on page 5, but only defined several pages later. Other abbreviations such as HR and AF are used throughout the manuscript - I supposed it's fairly standard in epidemiology that HR is used as an abbreviation for hazard ratio, but we have to wait until page 27 for it to be defined that way, and AF puzzled me throughout the manuscript until I came across an explanation on page 34 of the 35-page manuscript (apparently it stands for attributable fraction, but I still don't know how that's calculated).

6. The authors should be more explicit about how they created the dataset. Although I wasn't previously aware of it, I did consult the DHS website at <https://www.dhsprogram.com/> - it appears that one can register as a researcher and then download the individual-level datasets - I didn't try to do that myself, but this does seem to be the process. Is that correct, and if so, how exactly did the authors construct the dataset? - it would not harm to give the exact code they used. Similarly, the other datasets need to be more carefully documented.

7. p. 7: one question I have about statements like "MEI at the 90th percentile of exposure" (line 132 here, but there are similar statements at many other points in the paper) is to understand better what they mean by this kind of comparison. On the face of it, the statement means that a group of children exposed at the 90% level had a higher mortality rate than another group of children at the reference level mentioned on line 131 - in other words, it's written as if it was directly comparing two groups. I don't THINK that's what these authors mean. I think what they mean is that they fitted the nonlinear regression by a 3-DF spline relationship as described later in the paper, and then used the parametric form of spline function to compare the fitted response at the 90% level with that at the reference level, together with a side calculation of the standard error (but the statement is based on the entire dataset, not just individuals at either the 90% or the reference level of exposure). I would like the authors to clarify this point.

Minor points on specific textual issues:

8. p. 5, "disproportionally tele-connect regional and local weather conditions" - what does this mean, and how does it affect the analysis in this paper? Isn't this just a roundabout way of saying that El Niño events affect local weather conditions, or is there a specific reason for bringing out the "teleconnection" aspect of this?

9. p. 9, line 172 - I'm assuming "NESO" was just a typo for "ENSO"

10. p. 11, lines 220-221: again the authors used the word "teleconnect" without explaining what they mean by this (or what is their point about "variations in ENSO could disproportionately teleconnect regional weather")

11. p. 12, line 241: first mention of "extreme temperature" but shouldn't this have been mentioned earlier?

12. Bottom of p. 12: "Using extended Cox regressions coupling with time-dependent variable..." - again, the authors need to be more explicit about the statistical analysis they are doing.

13. p. 13, the fact that the focus on LMICs "might limit the generalizability of the results to individuals who were from more developed regions". This seems to me a very odd comment. The whole point of the paper is to examine climate effects in less-developed countries - why even speculate about whether these effects exist in developed countries (which could be the focus of some other study using completely different data)?

14. p. 32, bottom - "distributed lag non-linear models" - yet another place where the authors are completely vague about what they mean! A distributed lag model is one that fits a separate exposure effect to each possible lag (which, presumably, here refers to lags in months). A "distributed lag non-linear model", as I would understand it, means a separate non-linear function for each lag. But this implies a considerable increase in the number of estimated parameters and therefore the potential that the results might be affected by collinearity or other consequences of an overparameterized model. So yet again, I think we need more detail.

15. p. 34, line 543, I assume "sensitive" should be "sensitivity"

16. p. 35: "Supplementary information is available" - where?

Just for clarification, I did review the "Extended Data" supplement but I do not have a separate set of comments about that - I don't think it answers the numerous queries I have about the authors' statistical methods.

Overall: I feel this paper is potentially saying something important and therefore is worthy of consideration for Nature Communications, but there are a lot of problems with it at the moment. It's possible these queries could be cleared up with a suitable revision, but it would need to be a very thorough revision.

Reviewer #2 (Remarks to the Author):

The study found that maternal exposure to El Niño Southern Oscillation (ENSO) was associated with higher child mortality rates, especially under El Niño-like conditions. The study used a retrospective cohort design with a large sample of 1,678,065 children under five years of age from 38 low- and middle-income countries, based on cross-sectional data from 160 nationally representative Demographic and Health Surveys (DHS) conducted between 1981-2018.

The study provides valuable evidence for the association between ENSO and child mortality rates in low- and middle-income countries. However, there may be limitations in terms of sample size, representativeness, and unmeasured confounders that could affect the association between ENSO exposure and child mortality rates. Please consider the following and address or add accordingly to the limitations section:

- The study only focused on child mortality rates and did not investigate other health outcomes associated with ENSO exposure.
- Potential changes in ENSO exposure over time were not accounted for, which could affect the association between ENSO exposure and child mortality rates.
- The study relied on self-reported data for some variables, which may be subject to recall bias or misclassification.
- Potential differences in the association between ENSO exposure and child mortality rates across different regions or countries, age groups, and seasons were not investigated.
- The study did not investigate potential interactions between ENSO exposure and other environmental factors that may affect child health outcomes.

- The study's sample size, representativeness, and unmeasured confounders may affect the association between ENSO exposure and child mortality rates.
- The details of some aspects of the methodology, such as how missing data were handled or how variables were defined and measured, are unclear.

In addition, please consider the following suggestions to improve the structure of the paper:

- Please provide more details on certain aspects of the methodology, such as how missing data were handled or how variables were defined and measured.
- Avoid making claims such as "we, for the first time, investigated" and instead focus on the novelty or significance of the findings.
- Consider looking into the potential differences in the association between ENSO exposure and child mortality rates across different regions or countries, age groups, and seasons.
- Discuss the potential limitations of the study more thoroughly, including limitations related to sample size, representativeness, and unmeasured confounders as mentioned.
- Investigate potential interactions between ENSO exposure and other environmental factors that may affect child health outcomes.
- Provide more information on the statistical analysis methods used to allow others to fully reproduce their analysis.
- Add more details about how "Extreme ENSO events, including warm (El Niño-like) conditions, can affect human health" since the connection with health outcomes and why it leads to mortality should be clarified.
- Specify what specific changes the literature supports regarding the effects of climate change on maternal epigenetic changes and how these changes may relate to the association between ENSO exposure and child mortality rates.

Lastly, I would like to congratulate the authors for their efforts in conducting this study on the impact of maternal exposure to ENSO on under-five child mortality. The study's findings contribute to our understanding of environmental determinants of child mortality and highlight the need for further research in this area.

Reviewer #3 (Remarks to the Author):

This is an interesting paper that uses a large dataset of child health. Overall, the conclusions of the paper are limited by a lack of a clear hypothesis. A major limitation of the paper is the lack of understanding what climate variability is, how to describe ENSO and its indices, and how the components of climate variability are known to affect child mortality. There is considerable literature on this topic that is not referenced.

The exposures measure is not well defined. Maternal exposure to climate variability is not meaningful or useful. The title could more clearly describe the methods which is the quantification of the effect of El Niño indicators on annual child mortality.

The authors need to clarify which meteorological factors are driving this association.

The outcome data is not sufficiently described in detail. Is it the case the all deaths were pooled before the analysis? What is the justification for this? It may be that temperature is driving these associations- but this is not a new finding. The discussion section only describes some papers on ENSO and climate impacts on health but does not relate these to the findings. There is a lack of critical assessment of the methods.

The statistical methods need to be checked by a statistician.

There is no discussion of the impacts of ENSO in Africa and Asia. ENSO has very regional or localised impacts, that also vary from event to event. Overall the description of ENSO is very poor- it is a driver of interannual climate variability. Climate variability is a very general term - and it is also something every person on the planet experiences. The authors should report country level

results and discuss these in the context of known ENSO teleconnections. As ENSO has regional effects, there needs to some justification for the combined final estimate and what this actually means.

The methods are not clearly described. There are also some basic errors of interpretation. Maternal exposure included 12 months prior to birth, for example. ENSO is described as a modifiable risk factor.

The authors and editors of the journal should be mindful of the decolonisation of global health research. It is not acceptable to publish a study using African datasets without African authors who are experts in understanding the limitations of the health data, and also African climatologists who are expert in understanding the impacts of El Nino on the continent. The lack of a robust discussion on potential pathways means that implications of this finding are not related to actual policies and the current development of seasonal climate services for health.

Response to Reviewer Comments

We appreciate the thoughtful and valuable comments from three reviewers and the opportunity to extensively improve our manuscript, “Maternal Exposure to Climate Variability and Child Mortality: A Multi-country Study” (NCOMMS-23-11844-T), which has a slightly different title, based on these comments. Below, we have provided the point-by-point responses (in bold) to the reviewer’s comments (in plain text). Please note that the page numbers refer to the clean double-spaced version of the revised manuscript with updated analyses and discussions without “Track Changes” that we have submitted. References included in this document are to support point-by-point responses, as needed.

Reviewer #1 (Remarks to the Author):

The purpose of this paper is to examine the relationship between the meteorological phenomenon known as El Nino Southern Oscillation (ENSO), and childhood mortality in a selected set of 38 low and middle income countries (so-called LMICs). ENSO is measured using one of five indicators, abbreviated MEI, ESPI, ONI, Nino 1+2 and Nino 3.4. The outcome variable is one of total under-5 mortality (U5M), infant mortality (meaning mortality under one year of age) and neo-natal mortality (under one month of age), the latter two categories accounting for respectively 77% and 43% of the total U5M. Regression models were fitted to show the relationship between ENSO and mortality using non-linear splines and possibly some additional covariates (the last point is never made completely clear). In addition, the authors performed numerous subgroup analyses. The overall conclusion is that there is a statistically significant positive relationship between U5M and high levels of El Nino, typically characterized by the difference between the 90th percentile of El Nino and the zero value of the respective index. The authors' conclusion from this is the need for better early warning systems and mitigation measures during future El Nino events.

I think this paper has merit in collecting and analyzing a large dataset from the Demographic and Health Surveys (DHS) Program and relating it to numerous demographic and climate indicators. I had not previously heard of this dataset, though I presume that the present authors are not the first to have analyzed it to look for mortality patterns; that fact alone makes this seem to me a useful exercise. However, I have two major criticisms of what they do, and numerous more minor queries.

Response: We appreciate the reviewer’s recognition of the value of our research and insightful comments to help improve our work. We have tried our best to address the important points raised below. In particular, we have added a subsection titled “Covariates” under the “Methods” section to explain our regression model as clearly as possible (starting from line 671 on page 39). We have also added more details about the DHS data in the updated subsection “Study Design and Study Population” (from line 605 on page 34). Indeed, this DHS dataset has been used by a few researchers to look for mortality patterns in their exciting work, including one of our coauthors, who published his joint work on The Lancet in 2021, studying the effects of armed conflict on the health of women and children.¹ Details on how we address the queries raised by the reviewer are given below.

My major queries:

1. Why do they assume these mortality patterns are due specifically to El Nino and not just to the most familiar meteorological variables, temperature and precipitation? I am not very familiar with the literature on childhood mortality but for adult mortality (e.g. studies based on the US Medicare database, which is mostly adults over 65) there is a huge literature on climate-related mortality patterns, that generally supports the present authors' claim of a J-shaped curve (a non-linear relationship that curves sharply upwards at the high-temperature end of the curve but not at the low-temperature end). Without taking account of this very obvious confounder, it seems to me that claims of a causal effect are misguided.

Response: We thank the reviewer for raising this question. Climate variability, particularly that driven by ENSO, has been increasingly recognized as a crucial determinant of human health. In fact, ENSO, a periodic climatic phenomenon of variations in sea surface temperature, pressure, and wind in the equatorial Pacific Ocean, is a principal source of global climate variability and a driver of the local meteorological patterns (including temperature and precipitation patterns) in relatively remote regions through its tele-connection impacts. In our revised paper, we note that (from line 64 on page 3): “Two opposing climate patterns of ENSO, including warm (El Niño-like) and cold (La Niña-like) conditions, can heighten the probability of extreme weather events in certain regions...²”

Moreover, the tele-connection impacts of ENSO on local weather patterns are noted to be “predictable” (from line 70 on page 4): “El Niño events are observed in association with increased sea surface temperatures in western Indian Ocean, above-normal rainfall in Eastern Africa, and above-normal temperatures in parts of Southeast Asia.³”

Given the linkages between ENSO and local meteorological variables as well as those between local meteorological variables and human health, we hypothesize that ENSO might trigger a wide array of health risks, and investigate the linkage between a source of climate variability and human health directly. Thus, in our framework, mortality patterns may be due to ENSO *through* local meteorological variables, including the familiar ones such as temperature and precipitation. As a result, these familiar meteorological variables become potential “mediators” or “mechanisms” (rather than confounders).

One reason of directly associating ENSO with mortality patterns is practical: With several ENSO measures closely monitored, well predicted, and publicly available, the public and policymakers across the globe can benefit from understanding how

ENSO can be associated with population health directly. Further, the association between ENSO and mortality tends to reflect a cumulative and longer-term effect as it can be a driver of a series of local weather events (e.g., heat waves, cold spells, and droughts). This allows the global public health risks' "alert system" to put forward actions several months ahead of time.

Further, as we noted (from line 80 on page 4), a recent report from the World Health Organization (WHO) indicates that under-five mortality (U5M) can serve as a useful indicator for monitoring the overall impacts of ENSO events including El Niño. This is another reason why we associate ENSO with U5M.

2. I have really serious concerns about the way these authors present their statistical analysis, which go well beyond answering a few questions about fine-level details of the analysis. Throughout the paper, they are vague about what they are doing; they don't provide any of their computer code, nor do they give enough analytic detail for the reader to have any hope of reproducing their analysis. I already referred to the authors' vagueness about covariates; specifically, on page 33 they say "the following covariates were considered..." but what does "considered" mean? Were they included or not? If some were and some weren't, how did they decide which ones to include? There are standard statistical methods for deciding which covariates to include in a regression equation, from the very traditional (such as stepwise regression), through more modern methods such as ridge regression or lasso, up to machine learning techniques such as random forests or boosting - I am not necessarily implying that they should have used machine learning methods but they do need to say what they actually did. Other necessary details of the statistical analysis are only very loosely described, e.g. "quadratic B-splines with with three degrees of freedom" (what exactly does this mean? - either give a formula or a reference) or "corresponding cross-basis function with three knots at equally-space log scales" (same comment). I'm also concerned about the lack of any discussion of country by country effect (they do some subgroup analyses restricting attention to "geographic zones", but they don't consider country as a possible

covariate in their main analyses). This is of concern because one would expect mortality rates to differ substantially by country, and this is a possible confounder in an analysis like this.

Response: We thank the reviewer for these important comments. In the current version of our “Statistical Analysis” subsection, we added more details to make clear what we are doing, including the literature and main computer code (see lines 733-737 on pages 41-42 and lines 826-832 on page 46), where DLNM stands for “distributed lag non-linear models”:

“Consistent with previous studies, our Cox models included the cross-basis function of monthly ENSO measures built by the DLNM, which were specified by B-spline using the ‘bs’ function. Regarding the space of lags in the cross-basis function, three knots placed at equally-spaced log scales using the ‘lognots’ function.⁴⁻⁶” (*Note that, these updated sentences and the papers cited here explain the “B-spline” and “cross-basis function” we mentioned in the previous version of our manuscript.*)

“All data analyses were conducted between October 2022 and August 2023 and analyzed using R, version 3.6.3 (R Project for Statistical Computing), and the following packages were primarily used: package “survival” was installed to characterize associations with outcome measures; package ‘glmnet’ was loaded to select variables by LASSO; package ‘dlnm’ was applied to specify the cross-basis function for ENSO exposure and to predict and graphically plot the results of the fitted regression models.”

We hope that, with enough analytic details and the R codes (full replication codes available) provided, readers can find it easier to reproduce our analysis.

Regarding covariates, we have added a separate subsection titled “Covariates” to

explain in details the meaning of each covariate and how they are constructed. In the “Statistical Analysis” subsection, we further clarify how we “consider” these covariates (see lines 737-740 on page 42): “Because certain covariates might have unequal influences on the outcome measures, we used a Least Absolute Shrinkage and Selection Operator (LASSO) regression method to select the above-mentioned covariates for each metric of mortality separately.” More information on the modeling procedures of covariate selections is provided on lines 740-750 on page 42, and Extended Data Figure S5 graphically illustrates the selection of covariates to be included.

Finally, we added discussions of country-by-country effects based on our random-effects meta-analysis (see Figure 5, and Extended Data Figures S3 and S4) as well as the literature. In fact, in all of our primary analysis models, we have decided to always control for the country fixed effects as confounders—thus, we do not rely on the LASSO approach to decide when to include the country fixed effects. Then, in the first step of our meta-analysis, we found that (see page 10’s lines 196-198 and 199-200):

“...moderate to high levels of heterogeneity was observed for the associations between mothers’ exposure to selected ENSO measures (e.g., MEI) and child mortality across countries” and countries “more intensely affected by high levels of ENSO were Kenya, Rwanda, Tanzania, and Uganda, Madagascar, Malawi, and Zimbabwe.”

In the updated “Discussion” section, on lines 301-303 of page 15, we further note that “the WHO has identified several high-risk countries in Eastern Africa and recommended some close-monitoring countries in Southern Africa.⁷” These countries that could potentially receive more El Niño climatic impacts were in line with our country-specific findings.

Other queries that are less important but I would still like to see answered:

3. Why did the authors consider five different measures of El Nino and what additional information do they think they gained from doing so (as compared with the alternative strategy of just using one measure throughout the analysis)? I am aware that different meteorology research groups have different preferences on this point, and that sometimes they may prefer one measure over another based on which region of the earth they are studying, but it's not clear to me why you would want to repeat the analysis five times over with different measures. Moreover, extended Figure S1 seems to show that they all go up and down more or less together, so why make a distinction?

Response: We thank the reviewer for the comment. ENSO is a large, complex, and dynamic system, involving different aspects of the ocean and the atmosphere over the tropical Pacific Ocean. Using several different indexes derived from various measurements at sea level, such as sea surface temperature and pressure, instead of a single index, would be informative and beneficial in measuring the overall ENSO state.

In our supplementary materials, detailed information on the time series of each ENSO indicator was summarized in Extended Data Table S7, which might provide more background information for the readers to understand our findings. However, it might not be appropriate for us to jump into any conclusion on which indicator is better, and it could also be misleading if we prioritized one indicator over another. Thus, we have shown the different shapes of the association between different ENSO measures and health outcomes in the manuscript to give readers a more complete picture of the relationship.

Although Extended Data Figure S1 shows that the five measures of ENSO all go up and down more or less together, the tele-connection impacts of ENSO could be sensitive to a slight deviation of its measure. Thus, using several measures could

also be regarded as a robustness check. Besides, from a practical perspective, the public and policymakers could decide how likely they are facing climate threats, based on the number of indicators that are associated with mortality risks in their areas.

4. Another concern I have is the use they make of different lags in the ENSO values. I am well aware of and generally support the need to consider lags in any analysis of this nature, because in general there are lags of between a few months up to at least a year between the ENSO signal and its effects, but one has to be very careful to handle the lags in a way that does not raise suspicions of cherry-picking your results. Despite the authors' claims, I was unable to perceive a clear pattern in the various panels of Figure 3. I believe they could have taken a more systematic approach to the selection of suitable lags that would avoid the concern about selecting lags to fit their hypothesis.

Response: Thanks for this kind reminder. We would like to emphasize that, for all regressions, we included all lags from the current month up to 12 months, rather than only some of them. Our models estimate the individual as well as cumulative effects for all these lags. Therefore, we have tried our best to avoid any “cherry-picking”.

After updating our regressions (with a LASSO approach to select covariates), Figure 3 presents a much clearer pattern. As stated on lines 146-149 of page 8: “Overall, we found that extremely high levels of ENSO measures in multiple periods, and particularly the 0th-1st and 6th-12th lagged months were associated with an increased risk of child mortality.” This pattern is now quite robust across the five measures of ENSO we select (including MEI, ESPI, ONI, Niño 1+2, and Niño 3.4). These time windows correspond to the last month before delivery and the few months both before and after conception.

5. Just as a textual point, the authors need to be more careful about defining their

terminology as they go. For example, MEI and ONI are first mentioned on page 5, but only defined several pages later. Other abbreviations such as HR and AF are used throughout the manuscript - I supposed it's fairly standard in epidemiology that HR is used as an abbreviation for hazard ratio, but we have to wait until page 27 for it to be defined that way, and AF puzzled me throughout the manuscript until I came across an explanation on page 34 of the 35-page manuscript (apparently it stands for attributable fraction, but I still don't know how that's calculated).

Response: We are grateful for the reviewer's careful check of our manuscript! In our revised version, we have defined terminologies and abbreviations on their first appearance. As for attributable fraction (AF), we have added the details on how we calculate it on page 44 from line 789 to line 797:

“Then, regression coefficients corresponding to ENSO measures were used to calculate the AF of deaths and the number of deaths attributable to extreme climate variability, using commonly described approaches as follows:⁸

$$AF = \frac{HR - 1}{HR} \times 100$$

where HR is the hazard ratio of La Niña, weak El Niño, or moderate El Niño conditions considering the Neutral condition as the reference. We calculated the number of deaths attributable to exposure to La Niña, weak El Niño, or moderate El Niño by multiplying the AF with the number of deaths under these conditions.”

6. The authors should be more explicit about how they created the dataset. Although I wasn't previously aware of it, I did consult the DHS website at <https://www.dhsprogram.com/> - it appears that one can register as a researcher and then download the individual-level datasets - I didn't try to do that myself, but this does seem to be the process. Is that correct, and if so, how exactly did the authors construct the dataset? - it would not harm to give the exact code they used. Similarly, the other datasets need to be more carefully documented.

Response: We thank the reviewer for pointing out the need to be more explicit about how we created the dataset. In our “Study Design and Study Population” subsection right under the “Methods” section, we provided more details about the process of constructing our dataset.

For example, on page 37, we wrote: “In this study, we included all the publicly available standard surveys facilitated by IPUMS-DHS... IPUMS is an online data dissemination tool (<https://www.idhsdata.org/idhs/>) that has harmonized various variable names and codes across many DHS samples, making it easier to compile the relevant information on different types of data and perform comparative research comprehensively... As shown in Extended Data Tables S1 and S2, the enrolled countries could be further categorized into eight geographic regions based on the definition of the United Nations...” (lines 627-641). Then, on page 38, we further wrote: “Additionally, because the proportion of missing data for all covariates of potential interests in the remained observations was under 10% (Extended Data Table S3), we did not implement any imputation approach” (lines 651-654).

In fact, the IPUMS-DHS account is linked to the DHS account. One can register a user account, before downloading data from the website. We have provided some details about the process of getting access to the data on page 36: “DHS data are available free of charge from the program website (<https://dhsprogram.com/>) after researchers register a user account online, provide a brief description of the proposed projects, and receive the approval from the DHS Program” (lines 613-616). As noted, one needs to submit a research proposal to be reviewed by the DHS Program. The proposal should specify which datasets to be used and the Program will only grant the applicant the access to the specific datasets. One can, of course, apply for all available datasets at once.

As for other datasets used in this paper, we also try our best to document them more carefully, whenever necessary.

7. p. 7: one question I have about statements like "MEI at the 90th percentile of exposure" (line 132 here, but there are similar statements at many other points in the paper) is to understand better what they mean by this kind of comparison. On the face of it, the statement means that a group of children exposed at the 90% level had a higher mortality rate than another group of children at the reference level mentioned on line 131 - in other words, it's written as if it was directly comparing two groups. I don't THINK that's what these authors mean. I think what they mean is that they fitted the nonlinear regression by a 3-DF spline relationship as described later in the paper, and then used the parametric form of spline function to compare the fitted response at the 90% level with that at the reference level, together with a side calculation of the standard error (but the statement is based on the entire dataset, not just individuals at either the 90% or the reference level of exposure). I would like the authors to clarify this point.

Response: We confirm that it is indeed the latter meaning written by the reviewer. To clarify this point, we added the following sentence in the "Statistical Analysis" subsection on pages 45-46 from lines 819 to 824:

"According to the interval of exposure-response curves, we calculated the effect of La Niña-like conditions one year prior to birth on child mortality by comparing the 10th percentile (P10) of exposure level with the reference value (zero), and the effect of El Niño-like conditions was calculated by comparing the 90th percentile (P90) of exposure level with the reference value (zero)."

Minor points on specific textual issues:

8. p. 5, "disproportionally tele-connect regional and local weather conditions" - what

does this mean, and how does it affect the analysis in this paper? Isn't this just a roundabout way of saying that El Nino events affect local weather conditions, or is there a specific reason for bringing out the "teleconnection" aspect of this?

Response: We thank the reviewer for raising this point. The prefix “tele-” of the word emphasizes the fact that ENSO can affect the weather conditions of remote regions.

As noted on page 3, “ENSO is a periodic and irregular climatic phenomenon of variations in sea surface temperature, pressure, and wind in the equatorial Pacific Ocean that occurs every three to seven years and acts as the crucial driver of global climate variability.” Since most (of our study) countries are not really close to the equatorial Pacific Ocean, the word “tele-connection” is often used.

9. p. 9, line 172 - I'm assuming "NESO" was just a typo for "ENSO"

Response: Yes, it is a typo and we have corrected it.

10. p. 11, lines 220-221: again the authors used the word "teleconnect" without explaining what they mean by this (or what is their point about "variations in ENSO could disproportionately teleconnect regional weather")

Response: We have changed the word “tele-connect” to “affect”. We would like to suggest that local weather conditions in different areas could be affected by ENSO differently.

11. p. 12, line 241: first mention of "extreme temperature" but shouldn't this have been mentioned earlier?

Response: We have forgone the term “extreme temperature”. Moreover, in the

current version, “no significant modification effects were found across subgroups defined by temperature” (lines 177-178 of page 9), and we added the following in the current “Discussion” section (lines 349-362 on page 17):

“In the context of climate change, global temperature levels are boosted by extreme levels of ENSO, and Southern Africa have recorded El Niño event-associated extreme heat in 2015-2016.⁷ Indeed, pregnant women are typically at high risk of direct heat exposure due to hormonal changes during pregnancy.¹⁰ Also, abnormal temperature exposure during in-utero development has been found to be associated with elevated risks of low birth weight, preterm birth, and child death.¹¹ In this study, the associations of child mortality with the 90th percentile of several ENSO measures were slightly stronger in participants residing in areas with high levels of precipitation, but no significant differences were observed across subgroups defined by the levels of temperature, suggesting that the detrimental effects attributable to El Niño-like conditions might not be exclusively mediated by precipitation and temperature. Future investigations on the exact mechanism responsible for the linkage between El Niño and worsening child survival are urgently warranted.”

12. Bottom of p. 12: "Using extended Cox regressions coupling with time-dependent variable..." - again, the authors need to be more explicit about the statistical analysis they are doing.

Response: This sentence is now changed to “Using Cox regressions coupling with DLNM...” (see page 8’s line 367) We have also added more details about our Cox regressions in the “Statistical Analysis” subsection (see its second paragraph):

“Associations between mothers’ *preconceptional and prenatal* exposure to ENSO and child mortality were evaluated by Cox proportional hazard models, with survival time calculated as the period from a child’s birth to the date of death or

the first day of the mother's interview... the Cox models incorporated with DLNM were therefore constructed...^{6,12}"

13. p. 13, the fact that the focus on LMICs "might limit the generalizability of the results to individuals who were from more developed regions". This seems to me a very odd comment. The whole point of the paper is to examine climate effects in less-developed countries - why even speculate about whether these effects exist in developed countries (which could be the focus of some other study using completely different data)?

Response: Thank you! This is a good point. Indeed, the point is to examine what happened to LMICs rather than developed regions, and might not be regarded as a limitation. We have removed this limitation from the current version and added some other limitations.

14. p. 32, bottom - "distributed lag non-linear models" - yet another place where the authors are completely vague about what they mean! A distributed lag model is one that fits a separate exposure effect to each possible lag (which, presumably, here refers to lags in months). A "distributed lag non-linear model", as I would understand it, means a separate non-linear function for each lag. But this implies a considerable increase in the number of estimated parameters and therefore the potential that the results might be affected by collinearity or other consequences of an overparameterized model. So yet again, I think we need more detail.

Response: We are really grateful for the reviewer's comment on this point. We have added more details on the DLNM, for example, on lines 733-737 of pages 41-42 (quoted above).

Moreover, to avoid overparameterization, we follow the reviewer's suggestion to adopt a LASSO approach for covariate selections, which has been described above.

15. p. 34, line 543, I assume "sensitive" should be "sensitivity"

Response: We thank the reviewer for the catch! In the current version, we have forgone the term “sensitive analysis” or “sensitivity analysis” but used the word “robustness”, as we have realized that our meta-analysis not only may serve as a sensitivity check but also can enable country-by-country discussions.

16. p. 35: "Supplementary information is available" - where?

Just for clarification, I did review the "Extended Data" supplement but I do not have a separate set of comments about that - I don't think it answers the numerous queries I have about the authors' statistical methods.

Response: The claim about “supplementary information” has been removed. All information (including that in the Extended Data) has been submitted. The queries the reviewer has about our statistical methods are now addressed in the “Methods” section at the end of the Manuscript file.

Overall: I feel this paper is potentially saying something important and therefore is worthy of consideration for Nature Communications, but there are a lot of problems with it at the moment. It's possible these queries could be cleared up with a suitable revision, but it would need to be a very thorough revision.

Response: We are again really grateful for the reviewer’s recognition of the value of our research and constructive comments to help upgrade our work. We have tried our best to address all the queries proposed above and we hope that after the thorough revision this research could be further considered. Thanks!

Reviewer #2 (Remarks to the Author):

The study found that maternal exposure to El Niño Southern Oscillation (ENSO) was associated with higher child mortality rates, especially under El Niño-like conditions. The study used a retrospective cohort design with a large sample of 1,678,065 children under five years of age from 38 low- and middle-income countries, based on cross-sectional data from 160 nationally representative Demographic and Health Surveys (DHS) conducted between 1981-2018.

The study provides valuable evidence for the association between ENSO and child mortality rates in low- and middle-income countries. However, there may be limitations in terms of sample size, representativeness, and unmeasured confounders that could affect the association between ENSO exposure and child mortality rates. Please consider the following and address or add accordingly to the limitations section:

Response: We really appreciate the reviewer's complement of our research and the following helpful comments. We have tried to address all the concerns raised below, and added some limitations accordingly.

- The study only focused on child mortality rates and did not investigate other health outcomes associated with ENSO exposure.

Response: We thank the reviewer for raising this point. To address this concern, we have provided and updated the rationale and motivation for why we focus on child mortality, especially under-five mortality (U5M), on pages 3 and 4's lines 80-82. To improve child survival worldwide, the Sustainable Development Goals have proposed a global target for all countries to end preventable deaths in under-five children by 2030. To this end, identifying potential health determinants of U5M are particularly essential for reducing it. So far, higher mortality risks have been linked to extreme levels of ENSO conditions among adults, but the evidence in

children has not been fully characterized. Admittedly, considering other outcomes (e.g., cause-specific mortality) also seem to be crucial and interesting; however, the necessary data were not available across most of the DHS samples. A recent report from the World Health Organization (WHO) thus suggests that U5M may serve as a useful indicator for monitoring the overall impacts of ENSO events including El Niño.⁷

Our collective findings provide evidence that climate variability originating from ENSO can be a determinant of child survival. In the current version, we have also pointed out the limitation of our current study and would like to suggest future research to look at health outcomes that have not yet been investigated by the literature, which may or may not lead to deaths right away (see lines 378 to 381 on page 18):

“Also, further investigations are warrant to assess the impact of ENSO exposure on the full-spectrum of health outcomes in children, thereby fully capture the detrimental health effects posed by ENSO.”

- Potential changes in ENSO exposure over time were not accounted for, which could affect the association between ENSO exposure and child mortality rates.

Response: We thank the reviewer for the comment. In fact, in the current model, the ENSO exposure is measured at each lagged month from 0 (the current month) to -12 (i.e., 12 months ago), which accounts for the dynamics of ENSO exposure at least partially. Also, as noted on page 41, associations “between mothers’ *preconceptional and prenatal* exposure to ENSO and child mortality were evaluated by Cox proportional hazard models, with survival time calculated as the period from a child’s birth to the date of death or the first day of the mother’s interview.” By controlling for the survival time, we hope that potential changes in ENSO exposure over time could be further accounted for.

- The study relied on self-reported data for some variables, which may be subject to recall bias or misclassification.

Response: We thank the reviewer for pointing out this limitation. We have added this limitation accordingly (see lines 375-377 of page 18): “Second, given that the information on some covariates were obtained from children’s mothers at the end of their interview, potential recall bias or misclassification might exist.”

- Potential differences in the association between ENSO exposure and child mortality rates across different regions or countries, age groups, and seasons were not investigated.

Response: To address this point, we have analyzed the potential differences in the association between our ENSO exposure and child mortality rates across different regions or countries, maternal age groups (as shown in Figure 4 and Figure 5, as well as Extended Data Figures S2 through S4). In the current version, we have further added the analyses by the birth quarters of children (Extended Data Table S4). Relevant interpretations of the findings have been added and updated in the “Discussion” section.

- The study did not investigate potential interactions between ENSO exposure and other environmental factors that may affect child health outcomes.

Response: In the revised paper, we have tried to explore the potential interactions between ENSO exposure and some environmental factors that could affect the health of under-five children. On page 43, we wrote: “Also, we assessed whether the associations with ENSO exposure might differ in participants exposed to different levels of weather conditions. Thus, we implemented subgroup analyses across tertiles of annual average ambient temperature and precipitation during

the investigation period, based on available data collected around each DHS cluster location (1,068,152 participants with temperature measurements and 1,108,695 participants with precipitation measurements). In addition, subgroup analyses were conducted by the type of fuel for cooking in the households based on data collected from 1,284,931 participants. Type of cooking fuels may serve as a proxy for indoor air pollution exposure and was classified into two categories according to the existing literature: solid fuel (e.g., use of wood, coal, grass, straw, charcoal, shrub, dung, or agricultural crop residues) versus clean fuel (e.g., use of electricity, natural gas, liquefied petroleum gas, biogas, and solar).¹³ In our updated analyses, we found that the associations with ESPI and Niño 1+2 were greater in areas with high levels of precipitation, but no significant modification effects were found across subgroups defined by temperature. Further, the deleterious effect of ENSO on U5M was lower for those who were from the households using clean fuels for cooking activities. The relevant results are provided in Extended Data Tables S4 and S5, and the differential association estimates across subgroups were tested using 2-sample z-tests as described on pages 43-44, between lines 778 and 784.

- The study's sample size, representativeness, and unmeasured confounders may affect the association between ENSO exposure and child mortality rates.

Response: Thanks for raising this point. We would like to say that the focus of the paper is to investigate the associations in LMICs. Nevertheless, we must admit that there are still some limitations regarding the representativeness and unmeasured confounders in the study. Therefore, we have added the limitation accordingly. For example, we have added the following sentences at the end of the “Discussion” section:

“Fourth, although this retrospective cohort study was performed by pooling analysis of more than one million individuals, which could yield sufficient

statistical power on assessing the overall relationship between ENSO exposure and child survival, given that we generated a good precision (narrow CIs) around the point estimates, it must be acknowledged that the study sample may not be representative of some conflict-affected areas in countries experiencing wars or insecurity, or countries that opt out of participating in the DHS.¹ The magnitude of this under-representativeness of conflict affected areas may vary from one country to another. For instance, in the Democratic Republic of the Congo (or Congo Democratic Republic), only 4 of the 540 sampled clusters were not accessible due to insecurity in the 2012/2014 DHS surveys (that is, less than 1% all sampled clusters were affected). Therefore, we speculated that conflict conditions were not likely to have substantial influences on the magnitude and direction of the associations observed in this study. Fifth, the detrimental mortality effects discovered from young children might limit the generalizability of the results to other age group (e.g., teenagers). Furthermore, we should also limit our interpretations of the results to regions shown in Figure 1. Notwithstanding, climate-related stresses can exacerbate health inequalities around the world, and participants residing in LMICs would bear a larger burden of ENSO exposure due to systemic economic injustices, such as poverty and poor environmental conditions and hygiene practices.¹⁴ We have recognized that this supposition should be tested in future investigations in more LMICs with ENSO tele-connected impacts.”

Regarding the reviewer’s concerns on unmeasured confounders, we agreed that we might not completely rule out the possibility of residual confounding posed by potentially unmeasured covariates and have added the relevant limitation on page 18’s lines 371-375. Nevertheless, we have controlled for a variety of crucial covariates based on the relevant biological mechanisms and existing literature. In our extensively revised manuscript, we have added a separate subsection titled “Covariates” to explain in details the meaning of each covariate and how they are constructed. In the “Statistical Analysis” subsection, we further clarify how we

“consider” these covariates (lines 737-740 on page 42): “Because certain covariates might have unequal influences on the outcome measures, we used a Least Absolute Shrinkage and Selection Operator (LASSO) regression method to select the above-mentioned covariates for each metric of mortality separately.” More information on the modeling procedures of covariate selections is provided on lines 738-748 on page 42, and Extended Data Figure S5 further provides a graphical illustration of the selection of covariates to be included.

- The details of some aspects of the methodology, such as how missing data were handled or how variables were defined and measured, are unclear.

Response: We thank the reviewer for this comment. In the revised manuscript, we have added several aspects of the methodology, including how we handle missing data—“because the proportion of missing data for all covariates of potential interests in the remained observations was under 10% (Extended Data Table S3), we did not implement any imputation approach” (lines 651-654 of page 38), and how variables were defined and measured—see the newly added subsection named “Covariates”.

After excluding individuals with missing data of all key variables, our primary analytic sample covered 1,499,727 children under five years of age.

In addition, please consider the following suggestions to improve the structure of the paper:

- Please provide more details on certain aspects of the methodology, such as how missing data were handled or how variables were defined and measured.

Response: We thank the reviewer for this suggestion. We have followed it and added more details.

- Avoid making claims such as "we, for the first time, investigated" and instead focus on the novelty or significance of the findings.

Response: Thanks a lot for the suggestion. We have avoided making such claims.

- Consider looking into the potential differences in the association between ENSO exposure and child mortality rates across different regions or countries, age groups, and seasons.

Response: As discussed above, in the revised paper, we have checked the potential differences in the association by region/country, maternal age group, and child birth quarter.

- Discuss the potential limitations of the study more thoroughly, including limitations related to sample size, representativeness, and unmeasured confounders as mentioned.

Response: We thank the reviewer for this crucial suggestion. As discussed above, we have followed the reviewer's advice to discuss the study's potential limitations more thoroughly.

- Investigate potential interactions between ENSO exposure and other environmental factors that may affect child health outcomes.

Response: We thank the reviewer for this instructive suggestion. As discussed above, we have investigated potential interactions between ENSO exposure and some environmental factors (e.g., temperature, precipitation, and type of cooking fuels as a proxy for indoor air pollution).

- Provide more information on the statistical analysis methods used to allow others to fully reproduce their analysis.

Response: We thank the reviewer for the comment. In the revised “Statistical Analysis” subsection, we have added more details to make clear what we are doing, including the computer code. We hope that, with enough analytic details and the R codes (full replication codes available) provided, readers can find it easier to reproduce our analysis.

- Add more details about how “Extreme ENSO events, including warm (El Niño-like) conditions, can affect human health” since the connection with health outcomes and why it leads to mortality should be clarified.

Response: We appreciate the reviewer’s comment. In our updated “Discussion” section, we have added substantial contents about the mechanisms of how extreme ENSO events can affect human health. For more details, please refer to lines 320 to 362 on pages 16-17.

- Specify what specific changes the literature supports regarding the effects of climate change on maternal epigenetic changes and how these changes may relate to the association between ENSO exposure and child mortality rates.

Response: We thank the reviewer for this comment. We have realized that using the term “epigenetic changes” (which lacked the supports from literature) might not be appropriate and thus have changed it to “malnutrition”.

In our updated “Discussion” section, we also have cited papers that support the effects of climate change on both mothers and fetuses through nutritional changes. Some discussions are quoted as follows (see more details from line 323 to 346 on pages 16-17): “... The WHO has proposed that water-borne diseases diarrhea, cholera, and malnutrition are among the most climate change-sensitive health responses.¹⁰ ... Life-threatening manifestations of diarrhea during pregnancy

include dehydration and undernutrition, which are due to inhibiting nutrient absorption in gastrointestinal tracts, contributing to weakened immune systems in both mothers and fetuses.^{15,16} El Niño-induced malnutrition might be a consequence of El Niño's impacts on both food security and elevations of diarrheal diseases.¹⁷ Vulnerable individuals such as women and their infants in marginalized communities may be at higher risks.¹⁸ A study reported that children born during and after the 1997-1998 El Niño event experienced negative impacts on their long-term nutritional status, characterized by shorter height and less lean mass.¹⁹ Further, the warmer El Niño conditions in 2015 were observed in associated with worsening of child undernutrition in most of the LMICs, with nearly six million additional children having lower weight during the El Niño compared to if El Niño was not occurred.²⁰ Growing studies have shown that maternal undernutrition is associated with low birth weight of children, and underweight is in turn an important predictor for child mortality.²¹”

Lastly, I would like to congratulate the authors for their efforts in conducting this study on the impact of maternal exposure to ENSO on under-five child mortality. The study's findings contribute to our understanding of environmental determinants of child mortality and highlight the need for further research in this area.

Response: We are very grateful for the reviewer's encouragement and helpful comments. We have tried our best to address the points raised above. We hope that after the revision our research could be further considered. Thanks!

Reviewer #3 (Remarks to the Author):

This is an interesting paper that uses a large dataset of child health. Overall, the conclusions of the paper are limited by a lack of a clear hypothesis. A major limitation of the paper is the lack of understanding what climate variability is, how to describe ENSO and its indices, and how the components of climate variability are known to affect child mortality. There is considerable literature on this topic that is not referenced.

Response: We thank the reviewer for recognizing the value of our research and providing the helpful comments below. We have tried our best to address the issues raised by the reviewer. In particular, we have added a clearer hypothesis (i.e., on lines 89-91 of page 5, we wrote “... we hypothesized that climate variability driven by ENSO would heighten child mortality risks in a relatively large geographic area that has often been regarded as tele-connected to ENSO.”). For clarity, the following explanations regarding the climate variability driven by ENSO and its tele-connected impacts have also been added and updated in our revised paper on pages 3-4’s lines 61-70 as: “ENSO is a periodic and irregular climatic phenomenon of variations in sea surface temperature, pressure, and wind in the equatorial Pacific Ocean that occurs every three to seven years and acts as the crucial driver of global climate variability.⁹ Two opposing climate patterns of ENSO ... can heighten the probability of extreme weather events in certain regions, which might trigger a wide array of health risks.² It has been shown that the magnitude of the impacts attributable to ENSO vary across regions, depending on how intensely El Niño/La Niña-like conditions affect local weather patterns (e.g., heat waves, cold spells, and droughts) of an area (namely tele-connected impacts).²²” Note that, detailed information on the definition of each ENSO indicator has been provided in Extended Data Table S7.

Additionally, we have largely extended our discussions on the components of “climate variability” by discussing potential mechanisms of how ENSO affects the

local weather conditions in remote areas and then leads to higher mortality risks in under-five children, which is supported by several newly added references from the literature, as described on pages 16-17's lines 320 to 362.

We have tried to address all concerns raised below, and added some limitations accordingly.

The exposures measure is not well defined. Maternal exposure to climate variability is not meaningful or useful. The title could more clearly describe the methods which is the quantification of the effect of El Nino indicators on annual child mortality.

Response: Thanks for this comment. We would like to clarify that our exposures include both maternal and in-utero exposures, as we measured the exposures one year before a mother gave birth to a child, during which the child experienced roughly 80% of the exposures with the mother. Moreover, the other "20% of the year" right before conception could be critical for a mother's future child's ability to survive, as indicated by our Figure 3. Moreover, it is important to note that, our investigations are conducted *at the individual level*, where the association between our ENSO exposure and child mortality was evaluated by a Cox proportional hazard model, with the survival time calculated as the period from each child's birth to the date of death or the first day of the (mother's) interview. Therefore, to be clearer, we have followed the reviewer's idea and the literature and changed the title of our paper to:

"Maternal *Preconceptional and Prenatal* Exposure to Climate Variability driven by El Niño Southern Oscillation and Child Mortality: A Multi-country Study"

The authors need to clarify which meteorological factors are driving this association.

Response: We thank the reviewer for this instructive suggestion. In the revised

manuscript, we have examined whether meteorological parameters (including temperature and precipitation) may serve as potential factors that drive this association. As shown in Extended Data Table S5, the associations with ENSO measures (particularly ESPI and Niño 1+2) were greater in areas with high levels of precipitation, but no significant modification effects were found across subgroups defined by temperature. Our findings indicate that, there could still be other meteorological factors driving the observed associations, and we suggest on lines 360-361 of page 17 that future “investigations on the exact mechanism responsible for the linkage between El Niño and worsening child survival are urgently warranted.” Nevertheless, we have proposed several mechanisms after consulting the existing literature in the updated “Discussion” section.

The outcome data is not sufficiently described in detail. Is it the case the all deaths were pooled before the analysis? What is the justification for this? It may be that temperature is driving these associations- but this is not a new finding. The discussion section only describes some papers on ENSO and climate impacts on health but does not relate these to the findings. There is a lack of critical assessment of the methods.

Response: We thank the reviewer for these great comments.

In the revision, we have added more details about the outcome data from DHS. For example, in the “Study Design and Study Population” subsection under the “Methods” section, we have added some descriptions about the measurements of outcome data on lines 618-625 of page 36: “... nearly all DHS samples were representative at the national level. Reproductive-age women within per primary sampling unit (namely a DHS cluster location, with approximately 20-30 households) were randomly selected using a two-stage cluster sampling design, which employed enrollment processes stratified by geographical region and urban/rural location. All DHS respondents answered a suite of general questions about household- and individual-level covariates and each child’s records at birth

or the time of interview no matter if the child was still alive.” Based on the micro-data, we conducted investigations *at the individual level* to explore the relationships between our ENSO exposure and child mortality using Cox proportional hazard models, with the survival time calculated as the period from each child’s birth to the date of death or the first day of the mother’s interview.

We have also tried to check if temperature and precipitation drive the associations between mothers’ *preconceptional and prenatal* ENSO exposure and child mortality. For example, on lines 174-178 of page 9, we have added: “Additionally, subgroup analyses were performed to examine the associations between ENSO and mortality across tertiles of meteorological parameters at each DHS cluster location. As shown in Extended Data Table S5, the associations with ESPI and Niño 1+2 were greater in areas with high levels of precipitation, but no significant modification effects were found across subgroups defined by temperature levels.” These results collectively suggested that the child survival worsened by El Niño-like conditions might not be exclusively mediated through changes in meteorologic factors, such as precipitation and temperature.

In our updated “Discussion” section, we have tried to propose several pathways that may explain the findings observed in this study based on the existing literature, and the following text was added between lines 320 and 362 (of pages 16-17) as:

“Various mechanisms have been implicated in the pathogenesis of child mortality risks associated with exposure to climate-related environmental stressors.^{7,22} El Niño conditions may enhance the probability of droughts, floods, fires, and heatwaves, all of which are deleterious to child survival.²³ The WHO has proposed that water-borne diseases diarrhea, cholera, and malnutrition are among the most climate change-sensitive health responses.¹⁰ Drier-than-normal conditions, flooding, and above-normal rainfall are capable of restricting the access to safe water, and flooded sanitation infrastructure can result in water-borne diseases.²²

It has been shown that El Niño is linked outbreaks of cholera and other diarrheal diseases possibility through flooding-related contaminated water.⁷ Likewise, the relationships between El Niño events and vector borne diseases (e.g., malaria) have been well corroborated in Africa and parts of Asia.²³ El Niño is also expected to change in dynamics of weather patterns, thereby heightening risks of gastrointestinal infections posed by bacterial pathogens in water.²⁴ Life-threatening manifestations of diarrhea during pregnancy include dehydration and undernutrition, which are due to inhibiting nutrient absorption in gastrointestinal tracts, contributing to weakened immune systems in both mothers and fetuses.^{15,16} El Niño-induced malnutrition might be a consequence of El Niño's impacts on both food security and elevations of diarrheal diseases.¹⁷ Vulnerable individuals such as women and their infants in marginalized communities may be at higher risks.¹⁸ A study reported that children born during and after the 1997-1998 El Niño event experienced negative impacts on their long-term nutritional status, characterized by shorter height and less lean mass.¹⁹ Further, the warmer El Niño conditions in 2015 were observed in associated with worsening of child undernutrition in most of the LMICs, with nearly six million additional children having lower weight during the El Niño compared to if El Niño was not occurred.²⁰ Growing studies have shown that maternal undernutrition is associated with low birth weight of children, and underweight is in turn an important predictor for child mortality.²¹ Additionally, both hot and dry conditions can trigger heat waves, wildfire smoke, poor air quality, consequently causing exhaustion cardiorespiratory diseases, and heat stroke as well as exacerbating underlying illnesses resulting in premature death.^{7,25} In the context of climate change, global temperature levels are boosted by extreme levels of ENSO, and Southern Africa have recorded El Niño event-associated extreme heat in 2015-2016.⁷ Indeed, pregnant women are typically at high risk of direct heat exposure due to hormonal changes during pregnancy.¹⁰ Also, abnormal temperature exposure during in-utero development has been found to be associated with elevated risks of low birth weight, preterm birth, and child death.¹¹ In this study, the associations of child

mortality with the 90th percentile of several ENSO measures were slightly stronger in participants residing in areas with high levels of precipitation, but no significant differences were observed across subgroups defined by the levels of temperature, suggesting that the detrimental effects attributable to El Niño-like conditions might not be exclusively mediated by precipitation and temperature. Future investigations on the exact mechanism responsible for the linkage between El Niño and worsening child survival are urgently warranted.”

Finally, we have added some assessments of our methods on pages 11-12 (see lines 221-252: “ Although the associations between ENSO exposure and child survival had not been systematically assessed, a cross-sectional study indicated that occurrences of under-two child mortality peaked during El Niño events.²⁶ Nevertheless, whether these phenomena could be explained by variations in the intensity of ENSO metrics warrant further investigations, and well-documented exposure-response relationships remain largely sparse. To our knowledge, prior quantification studies of the ENSO effects on mortality have generally focused on the adult population.^{6,27} Results from the annual aggregate health data of older adults living in North America indicated that significant increases in risks of annual morbidity and mortality rates were observed in association with even weak to moderate El Niño events among participants from the western United States, although no effects were observed in those residing in the southern United States and Canada.²⁷ One possible explanation of the findings is that the annual aggregation of death data irrespective of known ENSO-associated dynamic climate anomalies might have limited the ability to capture the ENSO’s teleconnection impacts. A recent investigation at the individual level that linked ENSO with a variety of health data in Chinese elderly showed that both El Niño and La Niña-like conditions were closely correlated with mortality risks, and U-shaped relationships were found in their analysis based on recent advances in statistical methods such as distributed lag non-linear models (DLNM).⁶ They also found that some individuals might be more prone to the mortality effects attributable to El

Niño-like conditions. Using DLNM allowed them to quantify a comprehensive time-course picture regarding the exposure-response relationship.¹² Herein, our analytical approach introduced the similar models to associate U5M with ENSO in a retrospective cohort study of young children with adjustments for crucial confounding variables. We observed approximately J-shaped associations between maternal ENSO exposure over the past year prior to delivery and child mortality, with stronger impacts attributable to more El Niño-like conditions. The findings were also confirmed by the regressions of ENSO measures as categorical variables. Moderate El Niño showed the strongest associations with child mortality, followed by weak El Niño, whereas La Niña exhibited null or negative associations when compared with the neutral conditions. Collectively, the current study provides evidence that inter-annual variability of climate driven by extreme levels of ENSO can be a trigger of worsening child survival.”

The statistical methods need to be checked by a statistician.

Response: We have consulted several publications that utilized the methods used in this paper, i.e., “Cox models incorporated with DLNM”.^{6,12} We have also added more details about our statistical methods in the revised version to give readers a way to replicate our paper. The full replication codes are ready to share.

There is no discussion of the impacts of ENSO in Africa and Asia. ENSO has very regional or localised impacts, that also vary from event to event. Overall the description of ENSO is very poor- it is a driver of interannual climate variability. Climate variability is a very general term - and it is also something every person on the planet experiences. The authors should report country level results and discuss these in the context of known ENSO teleconnections. As ENSO has regional effects, there needs to some justification for the combined final estimate and what this actually means.

Response: We thank the reviewer for these important suggestions! First, we have

added some discussions of the impacts of ENSO in Africa and Asia. For example, on lines 70-73 of page 4, we have added the following text: “El Niño events are observed in association with increased sea surface temperatures in western Indian Ocean, above-normal rainfall in Eastern Africa, and above-normal temperatures in parts of Southeast Asia.”³ Indeed, ENSO’s (tele-connections) impacts are quite regional and localized. Thus, in our current discussions, we are more specific about what types of climate variability are involved.

To be more specific, we now report country-level results (see the first step of our meta-analysis) and discuss the differential associations based on the known ENSO tele-connections. For example, from line 300 to line 318, we have added:

“More recently, given the consequences of El Niño between July and September in 2023, the WHO has identified several high-risk countries in Eastern Africa and recommended some close-monitoring countries in Southern Africa.⁷ The countries potentially receiving more El Niño climatic impacts were Kenya, Rwanda, Tanzania, and Uganda, Madagascar, Malawi, and Zimbabwe,⁷ which was consistent with our country-specific findings in the meta-analysis. It has been pointed out that, wet and dry conditions regulated by ENSO tele-connections may be partially responsible for the differential associations across the study areas.⁷ In Eastern Africa, El Niño is typically associated with higher-than-normal rainfall and increased risk of flooding, thereby contributing to river overflows downstream, population displacement, crop and livestock losses.²⁸ Indeed, positive correlations between selected ENSO measures (e.g., MEI) and local meteorological parameter precipitation were found in most areas of Eastern Africa in the present analysis. Further, water contamination caused by heavy rainfall and flooding could exacerbate and prolong the outbreaks of climate-sensitive diseases.²⁹ In contrast, El Niño events are associated with drier-than-normal conditions in Southern Africa, and significant food security impacts due to droughts-related reducing food production may occur in this region.^{30,31} All these El Niño climatic

impacts are likely key substrates for worsening maternal and fetal health, eventually leading to increasing the risk of child death.”

Finally, in terms of the combined effects, we are actually checking if the observed associations in some countries are still significant on a global scale, which could have global health implications. As shown in our meta-analysis, the countries with significant effects also receive higher weights, suggesting that they also cover more population and/or have more precise estimates. Therefore, the associations should receive attention from not only a few countries, but also the global community, due to their scope and scale, as well as the fact that the world is connected.

The methods are not clearly described. There are also some basic errors of interpretation. Maternal exposure included 12 months prior to birth, for example. ENSO is described as a modifiable risk factor.

Response: We thank the reviewer for this comment. We have extensively added the details of all our methods to be more transparent and avoid misleading words. The exposure is defined as an ENSO measure during the year before a mother’s delivery of (i.e., 12 months before giving birth to) a child, and our outcome is the child’s survival. For clarity, we have changed the term “maternal exposure” to mothers’ or maternal “*preconceptional and prenatal* exposure” throughout the manuscript. We realize that ENSO might not be a modifiable risk factor, and thus have forgone the word “modifiable” in the current version of the manuscript. To avoid any erroneous or unclear usages of words, we have carefully and thoroughly edited our manuscript.

The authors and editors of the journal should be mindful of the decolonisation of global health research. It is not acceptable to publish a study using African datasets without African authors who are experts in understanding the limitations of the health data, and also African climatologists who are expert in understanding the impacts of El Nino on

the continent. The lack of a robust discussion on potential pathways means that implications of this finding are not related to actual policies and the current development of seasonal climate services for health.

Response: We appreciate the reviewer’s reminder. To avoid the “decolonization” of global health research, we have invited a well-established scholar to coauthor this revised paper. He is from the Center for Tropical Diseases and Global Health and the Faculty of Medicine at Université Catholique de Bukavu in Democratic Republic of the Congo, which is located in one of our study countries in Africa. With his experience of publishing a paper using the DHS data on *The Lancet*, as well as his inputs into this current paper, we have come to realize the strengths as well as potential limitations of the health data provided by DHS and been more cautious in interpreting our results. Moreover, we have extensively cited related literature to support our discussions on the potential pathways through which ENSO affect child mortality in different areas, and most of them have been quoted above.

We sincerely hope that we have addressed the concerns raised by our reviewers and we could have our revised manuscript considered by *Nature Communications*. We are grateful for the opportunity to massively improve our paper, and we look forward to hearing more from the journal!

Reference

- 1 Bendavid, E. *et al.* The effects of armed conflict on the health of women and children. *Lancet (London, England)* **397**, 522-532 (2021). [https://doi.org/10.1016/s0140-6736\(21\)00131-8](https://doi.org/10.1016/s0140-6736(21)00131-8)
- 2 National Oceanic and Atmospheric Administration. What are El Niño and La Niña? <https://oceanservice.noaa.gov/facts/ninonina.html>.
- 3 Moore, S. M. *et al.* El Niño and the shifting geography of cholera in Africa. *Proc Natl Acad Sci U S A* **114**, 4436-4441 (2017). <https://doi.org/10.1073/pnas.1617218114>
- 4 Rodrigues, M., Santana, P. & Rocha, A. Modelling of Temperature-Attributable Mortality among the Elderly in Lisbon Metropolitan Area, Portugal: A Contribution to Local Strategy for Effective Prevention Plans. *Journal of urban health : bulletin of the New York Academy of Medicine* **98**, 516-531 (2021). <https://doi.org/10.1007/s11524-021-00536-z>
- 5 Chen, J. *et al.* Low ambient temperature and temperature drop between neighbouring days and acute aortic dissection: a case-crossover study. *European heart journal* **43**, 228-235 (2022). <https://doi.org/10.1093/eurheartj/ehab803>
- 6 Xu, H. *et al.* Associations of climate variability driven by El Niño-southern oscillation with excess mortality and related medical costs in Chinese elderly. *The Science of the total environment* **851**, 158196 (2022). <https://doi.org/10.1016/j.scitotenv.2022.158196>
- 7 World Health Organization. 2023. *Public Health Situation Analysis: El Niño*. <https://www.who.int/publications/m/item/public-health-situation-analysis--el-ni-o> (accessed 4 August 2023).
- 8 Yang, H. Y., Lee, J. K. W. & Chio, C. P. Extreme temperature increases the risk of stillbirth in the third trimester of pregnancy. *Scientific reports* **12**, 18474 (2022). <https://doi.org/10.1038/s41598-022-23155-3>
- 9 Power, S. *et al.* Decadal climate variability in the tropical Pacific: Characteristics, causes, predictability, and prospects. *Science (New York, N.Y.)*

- 374, eaay9165 (2021). <https://doi.org:10.1126/science.aay9165>
- 10 Rylander, C., Odland, J. & Sandanger, T. M. Climate change and the potential effects on maternal and pregnancy outcomes: an assessment of the most vulnerable--the mother, fetus, and newborn child. *Glob Health Action* **6**, 19538 (2013). <https://doi.org:10.3402/gha.v6i0.19538>
- 11 Pandipati, S. & Abel, D. E. Anticipated impacts of climate change on women's health: A background primer. *International journal of gynaecology and obstetrics: the official organ of the International Federation of Gynaecology and Obstetrics* **160**, 394-399 (2023). <https://doi.org:10.1002/ijgo.14393>
- 12 Gasparrini, A. Distributed Lag Linear and Non-Linear Models in R: The Package dlnm. *Journal of statistical software* **43**, 1-20 (2011).
- 13 Yu, K. *et al.* Association of Solid Fuel Use With Risk of Cardiovascular and All-Cause Mortality in Rural China. *Jama* **319**, 1351-1361 (2018). <https://doi.org:10.1001/jama.2018.2151>
- 14 Thomson, M. C. & Stanberry, L. R. Climate Change and Vectorborne Diseases. *The New England journal of medicine* **387**, 1969-1978 (2022). <https://doi.org:10.1056/NEJMra2200092>
- 15 Katona, P. & Katona-Apte, J. The interaction between nutrition and infection. *Clinical infectious diseases : an official publication of the Infectious Diseases Society of America* **46**, 1582-1588 (2008). <https://doi.org:10.1086/587658>
- 16 Martins, V. J. *et al.* Long-lasting effects of undernutrition. *International journal of environmental research and public health* **8**, 1817-1846 (2011). <https://doi.org:10.3390/ijerph8061817>
- 17 Patz, J. A., Campbell-Lendrum, D., Holloway, T. & Foley, J. A. Impact of regional climate change on human health. *Nature* **438**, 310-317 (2005). <https://doi.org:10.1038/nature04188>
- 18 Haines, A. & Ebi, K. The Imperative for Climate Action to Protect Health. *The New England journal of medicine* **380**, 263-273 (2019). <https://doi.org:10.1056/NEJMra1807873>
- 19 Danysh, H. E. *et al.* El Niño adversely affected childhood stature and lean mass

- in northern Peru. *Climate Change Responses* **1**, 7 (2014).
<https://doi.org/10.1186/s40665-014-0007-z>
- 20 Anttila-Hughes, J. K., Jina, A. S. & McCord, G. C. ENSO impacts child undernutrition in the global tropics. *Nature communications* **12**, 5785 (2021).
<https://doi.org/10.1038/s41467-021-26048-7>
- 21 Christian, P., Smith, E. R. & Zaidi, A. Addressing inequities in the global burden of maternal undernutrition: the role of targeting. *BMJ global health* **5**, e002186 (2020). <https://doi.org/10.1136/bmjgh-2019-002186>
- 22 World Meteorological Organization, 2016. *WHO-WMO Brief on Health and ENSO*. <https://public.wmo.int/en/media/news/who-wmo-brief-health-and-enso> (accessed 27 January 2016).
- 23 Anyamba, A. *et al.* Global Disease Outbreaks Associated with the 2015–2016 El Niño Event. *Scientific reports* **9**, 1930 (2019).
<https://doi.org/10.1038/s41598-018-38034-z>
- 24 Heaney, A. K., Shaman, J. & Alexander, K. A. El Niño-Southern oscillation and under-5 diarrhea in Botswana. *Nat Commun* **10**, 5798 (2019).
<https://doi.org/10.1038/s41467-019-13584-6>
- 25 World Health Organization. 2018. *Heat and Health*. <https://www.who.int/news-room/fact-sheets/detail/climate-change-heat-and-health> (accessed 1 June 2018).
- 26 Kaharuzza, F., Sabroe, S. & Scheutz, F. Determinants of child mortality in a rural Ugandan community. *East African medical journal* **78**, 630-635 (2001).
<https://doi.org/10.4314/eamj.v78i12.8931>
- 27 Majeed, H., Moineddin, R. & Booth, G. L. Sea surface temperature variability and ischemic heart disease outcomes among older adults. *Scientific reports* **11**, 3402 (2021). <https://doi.org/10.1038/s41598-021-83062-x>
- 28 Palmer, P. I. *et al.* Drivers and impacts of Eastern African rainfall variability. *Nature Reviews Earth & Environment* **4**, 254-270 (2023).
<https://doi.org/10.1038/s43017-023-00397-x>
- 29 Kovats, R. S., Bouma, M. J., Hajat, S., Worrall, E. & Haines, A. El Niño and

health. *Lancet* (London, England) **362**, 1481-1489 (2003).
[https://doi.org:10.1016/s0140-6736\(03\)14695-8](https://doi.org:10.1016/s0140-6736(03)14695-8)

30 Gore, M., Abiodun, B. J. & Kucharski, F. Understanding the influence of ENSO patterns on drought over southern Africa using SPEEDY. *Climate Dynamics* **54**, 307-327 (2020). <https://doi.org:10.1007/s00382-019-05002-w>

31 *United Nations Office for the Coordination of Humanitarian Affairs (UNOCHA)*. 2016. *El Niño: Overview of Impact, Projected Humanitarian Needs and Responses*. <https://reliefweb.int/report/world/el-ni-o-overview-impact-projected-humanitarian-needs-and-response-02-june-2016> (accessed 3 Jun 2016).

REVIEWER COMMENTS

Reviewer #1 (Remarks to the Author):

The paper is vastly improved and I have to commend the authors on the thoroughness with which they responded to comments (I didn't actually think they would go to the trouble of adding an African co-author, which was the comment of another reviewer, but they did; welcome, Dr. Malembaka!).

The authors have given considerably more detail about their data collection methods, statistical analyses, and computer code. On the latter point, I did check the github reference they provided and I can confirm that the code is there, but it is quite a short program (129 lines). Was this all the code they used? The question seems worth asking.

Beyond that, I think my only comments are minor textual issues.

l. 162: comma after "child-level"? (the comma after "mother-level" is the controversial "Oxford comma", which many English speakers including this one would omit, but I'll leave that to the authors and editors)

l. 180: this time "AF" is defined early on (thank you!) but the formula is only given much later (l. 793). I can see what the authors are doing, deferring all technical statements to the "Methods" section; it still seems odd to me that they leave this undefined when it first appears. Another issue for the authors and editors to sort out, I think.

l. 198: I'd like to see a reference for "I² statistics" (this also appears later, but is still undefined, I think)

l. 199-200: the grouping is a bit odd here (first Kenya, Rwanda, Tanzania, then "and", then Uganda, etc.). (See also line 304.) This suggests to me that there may have been some sub-grouping going on that is not explained, e.g. that Kenya, Rwanda and Tanzania were first grouped together and then analyzed jointly

l. 202: HRs? (Presumably just one, so "HR".) (Another comment I made previously is that I don't think the authors ever defined HR, and I would accept that "hazard ratio" is such a universal term in epidemiology that it could be stated without definition. Still, my own policy is, when in doubt, define your terminology.)

l. 267 "acknowledge" (no "d" on the end)? This is a very common typing error

l. 304: see comment on l. 199 above

l. 341-342: please proof read this section. The sentence straddling these two lines does not make sense which implies to me that something got dropped

l. 379 "warrant" -> "warranted"?

l. 380 "capture" -> "capturing"?

l. 388-393: the authors acknowledge that "underrepresentativeness" could be an issue, but then they only give one example, the Democratic Republic of the Congo, and state that it is not an issue there. So what about all the other countries in the survey? Now, I did notice that the new author, Dr. Malembaka, is from the Democratic Republic of the Congo, so maybe he had access to that country's data which could not be easily replicated in other countries. Still, if this was the reason for using a sample of size one, maybe there should be some acknowledgement of that fact (e.g. "we were unable to obtain comparable data for other countries")

l. 394-395: this raises a concern that I raised in a different context last time, that a phrase like "limit the generalizability" (in this case, to teenagers) reads rather oddly when the authors mean

very simply that they just didn't consider that population (and there's no reason why they should have, since the reasons for emphasizing early childhood mortality are very clearly stated early in the paper).

l. 744-745: slight repetition here: you don't need the first "cross-validated" when the next line says the result was "generated by cross-validation"

l. 811 (x-ref l. 198): please give a reference for "I-squared statistics"

I don't feel it should take the authors very long to address all these comments and then I'd be happy to see the paper proceed.

Reviewer #2 (Remarks to the Author):

In revisiting the revised manuscript, I appreciate the efforts made by the authors to address the comments and questions raised in my initial review. Your revised paper indeed incorporates critical changes that significantly improve its scientific merit and relevance.

Firstly, your updated rationale for focusing solely on child mortality rates in relation to El Niño Southern Oscillation (ENSO) is convincing. The alignment of your study with the United Nations' Sustainable Development Goals offers an essential contextual backdrop, and your acknowledgment of the necessity for future research in other health outcomes lends a balanced perspective to your work. Also, I commend your approach to accounting for changes in ENSO exposure over time through the use of lagged variables. Your methodological adjustment shows your commitment to rigor and serves to mitigate concerns regarding the impact of temporal variations in ENSO exposure on child mortality rates.

Furthermore, your acknowledgment of the limitations concerning self-reported data and recall bias strengthens the paper's integrity. While this limitation is inherent to many epidemiological studies that rely on self-reported measures, mentioning it provides the reader with a more nuanced understanding of the research. Moreover, the additional analyses investigating differences in child mortality rates across various strata, including regions, maternal age groups, and seasons, enrich the manuscript. This new information provides a more comprehensive picture of the nuanced effects of ENSO on child mortality across different subpopulations.

The inclusion of interaction terms with environmental variables is an important contribution to the paper. These new analyses introduce a depth of complexity and relevance that was missing in the initial version. Lastly, your acknowledgment of sample size, representativeness, and potential confounders contributes to a more rounded discussion of the study's limitations. Particularly, the mention of conflict-affected areas not covered in the Demographic and Health Surveys (DHS) is an insightful addition.

Overall, you have addressed each of the limitations and suggestions thoughtfully, substantively improving the manuscript. Based on these revisions, I have no further comments.

Reviewer #4 (Remarks to the Author):

I appreciate the authors' effort in such research and the rebuttal. However, still some major points that I think are fundamental to the quality of this work have not been addressed.

Major comments

1. General

a. Background introduction, terminologies, and methods were not clearly described. There were inconsistencies across different descriptions.

2. Research question: the authors are asking about exposure to climate variability. I understand that ENSO affects the global climate, and it is okay for authors to use ENSO to describe climate variability. My concern lies on the length of time window the authors selected, How could a one-year period show variability in climate (which has a much longer timespan)? To me, one-year

ENSO index is more likely to show extreme weather events and decades of ENSO index could represent the climate variability.

3. Descriptions on background

a. Lines 28-29: why jumped from infectious diseases to child mortality. I saw no clear description on the connection between these two.

b. Line 81: WHO lists more than 20 useful indicators, and U5M is just one of them. Please tone down here.

4. Methods

a. Seasonality of ENSO: considering the strong seasonality of ENSO, I think the stratified analysis by season of birth worth being included in the main text.

b. Use of lag 0-12 months: First, lag 0-12 months include 13 months, which is inconsistent with the description of "maternal ENSO exposure over the past year prior to delivery". At least, it is not clearly stated. Second, lag 0-12 months covered both pre-conception and pregnancy periods, the exposures of which periods could have very different impacts on the birth outcomes and fetus development. Figure 3 is not sufficient to justify putting these two periods together. Please conduct separate analyses for the two periods and provide more solid statistics to justify the use of lag 0-12 months.

c. Please clarify the reason for excluding neonatal mortality in stratified analysis.

d. Multiple comparison: The authors used factor-specific Bonferroni-corrected p-values for the effect modification analysis. However, the authors had no consideration of multiple factors (e.g., sex, birth order, mother age).

e. Country effects: Any specific reasons for controlling country effects instead of cluster effects?

f. Subgroup analyses by temperature and precipitation: As explained by the authors in the rebuttal, temperature and precipitation serve as mediators. Then how would authors interpret the ENSO estimates when stratified by mediators? Were they total effects of ENSO minus the effects through temperature/precipitation? Please be cautious here that the interpretation changed.

g. Classification of ENSO measures by 0.5 and 1: ENSO had associations with outcomes starting from values smaller than 0.5. Why did the authors still consider -0.5 to 0.5 as the reference group? How would that bias the estimates?

h. Calculation of attributable fractions: The formula is for RR, not HR. The authors should provide theoretical reasons in the extended data, why such an equation is applicable.

5. Results

a. Table 1: What is the meaning of numbers in parentheses?

b. Table 1: The way of showing the mother age group is not clear. For example, is 20 years old included in the first group or second group?

Minor comments

1. Line 666: "(death at <12 months of age", a "(" is missing.

2. Line 766: "Also, we assessed weather ..." should be "whether"?

Response to Reviewer Comments

We are extremely grateful for the thoughtful and valuable comments from the anonymous reviewers and the opportunity to further improve our manuscript, “Maternal Preconceptional and Prenatal Exposure to Climate Variability driven by El Niño Southern Oscillation and Child Mortality: A Multi-country Study” (NCOMMS-23-11844A), whose title has been slightly changed, based on these additional comments. Below, we have provided the point-by-point responses (in bold) to the reviewers’ comments (in plain text). Please note that the page numbers refer to the clean double-spaced version of the revised manuscript with updated analyses and discussions without “Track Changes” that we have submitted. References included in this document are to support point-by-point responses, as needed.

Reviewer #1 (Remarks to the Author):

The paper is vastly improved and I have to commend the authors on the thoroughness with which they responded to comments (I didn't actually think they would go to the trouble of adding an African co-author, which was the comment of another reviewer, but they did; welcome, Dr. Malembaka!). The authors have given considerably more detail about their data collection methods, statistical analyses, and computer code. On the latter point, I did check the github reference they provided and I can confirm that the code is there, but it is quite a short program (129 lines). Was this all the code they used? The question seems worth asking.

Response: We really appreciate your recognition of our efforts made in the revised manuscript. We have to thank you for all the suggestions provided to help us vastly improve the paper. Again, we have tried our best to address the additional points raised below. Moreover, we have uploaded the updated version of our codes with more complete details to GitHub.

Beyond that, I think my only comments are minor textual issues.

l. 162: comma after "child-level"? (the comma after "motherlevel" is the controversial "Oxford comma", which many English speakers including this one would omit, but I'll leave that to the authors and editors)

***Response:* Thank you for catching the missing comma! Indeed, we should add the comma after “child-level”. We have kept the “Oxford comma” after “mother-level” to ensure clarity and consistency (e.g., when the lists are complex). We checked recent publications in Nature Communications and find that the use of an “Oxford comma” is common. Nevertheless, we will be sure to follow the style guide of the journal.**

l. 180: this time "AF" is defined early on (thank you!) but the formula is only given much later (l. 793). I can see what the authors are doing, deferring all technical statements to the "Methods" section; it still seems odd to me that they leave this undefined when it first appears. Another issue for the authors and editors to sort out, I think.

***Response:* Thank you for raising this point! Although the formula is only provided in the Methods section at the end still, in this revision, when “AF” first appears, we give a non-technical explanation and remind readers that the formula will be provided later (page 9, lines 193-196).**

“Extended Data Table S6 provides the estimated attributable fraction (AF), a measure used to quantify the proportion of health outcome (e.g., child deaths) in the population that can be attributed to a specific exposure (e.g., a high ENSO level). The formula is provided in the Methods section.”

l. 198: I'd like to see a reference for " I^2 statistics" (this also appears later, but is still undefined, I think)

Response: Thanks for the suggestion! In this revision, we define “I-squared” when it first appears and provide citations immediately (page 10, lines 216-220).

“The I-squared (I^2) statistic, a commonly used heterogeneity measure, was applied to describe the percentage of variability in meta-analysis; an I^2 value of $\leq 25\%$ indicates low level of heterogeneity, a value around 50% indicates moderate level of heterogeneity, and a value near 75% or greater corresponds to high level of heterogeneity^{1,2}.”

l. 199-200: the grouping is a bit odd here (first Kenya, Rwanda, Tanzania, then "and", then Uganda, etc.). (See also line 304.) This suggests to me that there may have been some sub-grouping going on that is not explained, e.g. that Kenya, Rwanda and Tanzania were first grouped together and then analyzed jointly

Response: We thank the reviewer for pointing out this typo. In fact, there was no sub-grouping, and the “and” was redundant. We have removed this misleading “and” to avoid confusion.

l. 202: HRs? (Presumably just one, so "HR".) (Another comment I made previously is that I don't think the authors ever defined HR, and I would accept that "hazard ratio" is such a universal term in epidemiology that it could be stated without definition. Still, my own policy is, when in doubt, define your terminology.)

Response: We have followed your suggestion and removed the “s” after “HR”. We also provide a non-technical definition when it first appears.

“...the hazard ratio (HR), i.e., the risk at the specified level of exposure divided by the risk at the reference level...” (page 7, lines 146-148)

l. 267 "acknowledge" (no "d" on the end)? This is a very common typing error

Response: Thank you! We have followed the suggestion to remove the “d” after “acknowledge”.

l. 304: see comment on l. 199 above

Response: We have corrected this typo by removing this misleading “and” to avoid reader confusion.

l. 341-342: please proof read this section. The sentence straddling these two lines does not make sense which implies to me that something got dropped

Response: Thank you for pointing out this sentence and the corresponding section. After careful reading, we believe that this sentence and the corresponding paper was mis-cited. Thus, we removed this citation and added one sentence summarizing the logic behind the above discussion.

“This evidence suggests a potential link between El Niño and undernutrition among both mothers and infants, which may lead to child underweight and infections, increasing child mortality in the longer term.” (page 18, lines 375-377)

l. 379 "warrant" -> "warranted"?

Response: Thank you for catching this. We have corrected it.

l. 380 "capture" -> "capturing"?

Response: Thank you for catching this. We have changed it.

l. 388-393: the authors acknowledge that "underrepresentativeness" could be an issue, but then they only give one example, the Democratic Republic of the Congo, and state that it is not an issue there. So what about all the other countries in the survey? Now, I did notice that the new author, Dr. Malembaka, is from the Democratic Republic of the Congo, so maybe he had access to that country's data which could not be easily replicated in other countries. Still, if this was the reason for using a sample of size one, maybe there should be some acknowledgement of that fact (e.g. "we were unable to obtain comparable data for other countries")

***Response:* We appreciate your suggestion! In this revision, we have acknowledged the following in our discussion of limitations:**

“Noteworthy was that this speculation was derived based only on the example of the Democratic Republic of Congo, given that we were unable to obtain comparable data for other countries.” (page 20, lines 425-427)

l. 394-395: this raises a concern that I raised in a different context last time, that a phrase like "limit the generalizability" (in this case, to teenagers) reads rather oddly when the authors mean very simply that they just didn't consider that population (and there's no reason why they should have, since the reasons for emphasizing early childhood mortality are very clearly stated early in the paper).

***Response:* We appreciate this suggestion. Accordingly, in this revision, we have removed the phrase.**

l. 744-745: slight repetition here: you don't need the first "cross-validated" when the next line says the result was "generated by cross-validation"

***Response:* Thank you for catching this. We've removed the redundant phrase.**

l. 811 (x-ref l. 198): please give a reference for "I-squared statistics".

***Response:* Thanks for the suggestion. In our previous response, we have provided the added definition and references.**

I don't feel it should take the authors very long to address all these comments and then I'd be happy to see the paper proceed.

***Response:* Thank you so much for the encouragement and wonderful suggestions. In this revision, we have tried our best to address all the queries proposed above and we genuinely hope that this paper can proceed.**

Reviewer #2 (Remarks to the Author):

In revisiting the revised manuscript, I appreciate the efforts made by the authors to address the comments and questions raised in my initial review. Your revised paper indeed incorporates critical changes that significantly improve its scientific merit and relevance.

Firstly, your updated rationale for focusing solely on child mortality rates in relation to El Niño Southern Oscillation (ENSO) is convincing. The alignment of your study with the United Nations' Sustainable Development Goals offers an essential contextual backdrop, and your acknowledgment of the necessity for future research in other health outcomes lends a balanced perspective to your work. Also, I commend your approach to accounting for changes in ENSO exposure over time through the use of lagged variables. Your methodological adjustment shows your commitment to rigor and serves to mitigate concerns regarding the impact of temporal variations in ENSO exposure on child mortality rates.

Furthermore, your acknowledgment of the limitations concerning self-reported data and recall bias strengthens the paper's integrity. While this limitation is inherent to many epidemiological studies that rely on self-reported measures, mentioning it provides the reader with a more nuanced understanding of the research. Moreover, the additional analyses investigating differences in child mortality rates across various strata, including regions, maternal age groups, and seasons, enrich the manuscript. This new information provides a more comprehensive picture of the nuanced effects of ENSO on child mortality across different subpopulations.

The inclusion of interaction terms with environmental variables is an important contribution to the paper. These new analyses introduce a depth of complexity and relevance that was missing in the initial version. Lastly, your acknowledgment of sample size, representativeness, and potential confounders contributes to a more rounded discussion of the study's limitations. Particularly, the mention of conflict-

affected areas not covered in the Demographic and Health Surveys (DHS) is an insightful addition. Overall, you have addressed each of the limitations and suggestions thoughtfully, substantively improving the manuscript. Based on these revisions, I have no further comments.

Response: We would like to express our greatest gratitude for your encouragement and helpful suggestions given before, which helped us tremendously in improving this work. We thank you for recognizing the efforts made in the revision as well as the values in our research.

Reviewer #4 (Remarks to the Author):

I appreciate the authors' effort in such research and the rebuttal. However, still some major points that I think are fundamental to the quality of this work have not been addressed.

Response: Thank you for commending us on the efforts in the previous round of revision and further providing us with more invaluable suggestions. We have tried our best to address the additional issues raised below.

Major comments

1. General

a. Background introduction, terminologies, and methods were not clearly described. There were inconsistencies across different descriptions.

Response: We thank the reviewer for raising this concern. To make sure that we clearly describe the background introduction, terminologies, and methods, and make consistent descriptions throughout the paper, we proofread the whole paper again and made necessary edits. For example, in the first paragraph, we have rephrased the motivation for conducting this research to make it clearer. Then, in the main text, we added the following sentence in the second paragraph (page 4, lines 68-69): “An episode of El Niño or La Niña generally lasts nine to 12 months, although it can sometimes last for years.³” This statement aims to justify our focus on the 0-12 lagged months prior to delivery. Moreover, in lines 75-83, we added more background information and justification for focusing on child mortality and the chosen exposure period:

“Pregnancy is perceived as a physiological ‘stress test’ because the maternal body is challenged with a variety of adaptive changes in cardiorespiratory, endocrine, and immune function.⁴ These maternal responses, along with the dynamic alterations in between fetoplacental circulation and rapid growth of fetus, may

confer both mothers and children more susceptible to the adverse effects of climate-related stresses.⁴ Further, the effects of climate-related stresses may further exacerbate the intergenerational cycle of malnutrition.^{3,5} Thus, it is plausible that ENSO anomalies may worsen maternal and fetal health, thereby leading to a heightened risk of child deaths.”

We have also added definitions of terminologies when they first appear, following the suggestions given by the first reviewer. We have added new details in the methods section following your suggestions below. Most of the other edits have been reflected in the responses below.

2. Research question:

the authors are asking about exposure to climate variability. I understand that ENSO affects the global climate, and it is okay for authors to use ENSO to describe climate variability. My concern lies on the length of time window the authors selected, How could a one-year period show variability in climate (which has a much longer timespan)? To me, one-year ENSO index is more likely to show extreme weather events and decades of ENSO index could represent the climate variability.

Response: We thank you for this important comment! Indeed, after carefully re-thinking about the appropriate usage of this terminology, we have decided to abandon the inappropriate use of the term “climate variability” in our paper. The title has been changed to “...Exposure to El Niño Southern Oscillation Levels...” rather than “...Exposure to Climate Variability...” as a result, and we have also replaced “climate variability” with “climate anomalies” or simply delete this term and refer the exposure to the level of ENSO, whenever necessary and appropriate. The focus on the one-year window is explained above, and we agree that it mainly reflects extreme weather events during the preconceptional and prenatal periods.

3. Descriptions on background

a. Lines 28-29: why jumped from infectious diseases to child mortality. I saw no clear description on the connection between these two.

Response: Thank you pointing this out. We have rephrased the sentences in these lines to provide a connection between switching from infectious disease to child mortality (page 2, lines 26-29):

“El Niño Southern Oscillation (ENSO) in the equatorial Pacific Ocean is a ubiquitous driver of global climate anomalies that have been shown to be associated with the epidemiology of childhood infectious diseases,⁶ but evidence for other aspects of child health and whether they increase child deaths is limited.”

Note that, child mortality is closely related to infectious diseases, but it contains other aspects of child health as well.

b. Line 81: WHO lists more than 20 useful indicators, and U5M is just one of them. Please tone down here.

Response: Thank you for the kind suggestion. We have followed your advice to tone down the statement by emphasizing that U5M is just “one of the useful indicators”.

4. Methods

a. Seasonality of ENSO: considering the strong seasonality of ENSO, I think the stratified analysis by season of birth worth being included in the main text.

Response: Thank you for the suggestion! We have followed your advice to move the stratified analysis by season of birth to the main text. To be more specific, in the updated Figure 4, we have added the results for different birth quarters right after those for child delivery places. We did not find consistent patterns of effect

modification by season of birth. For example, while Q4 is the birth quarter with the highest HR based on MEI, it becomes the one with the lowest HR based on ONI and Nino 3.4; Q2 becomes the one with the highest HR based on Nino 1+2. While the conclusion has been stated in the main text, these details are provided in the Extended Data file.

b. Use of lag 0-12 months: First, lag 0-12 months include 13 months, which is inconsistent with the description of “maternal ENSO exposure over the past year prior to delivery”. At least, it is not clearly stated. Second, lag 0-12 months covered both pre-conception and pregnancy periods, the exposures of which periods could have very different impacts on the birth outcomes and fetus development.

Figure 3 is not sufficient to justify putting these two periods together. Please conduct separate analyses for the two periods and provide more solid statistics to justify the use of lag 0-12 months.

***Response:* Thank you for pointing out this issue. We have corrected our description by removing the phrase “over the (past) year”, and write out the exact exposure window explicitly. We use “during and prior to delivery” occasionally to indicate the inclusion of the 0th lagged month. In the above responses, we have provided one justification for focusing on this 13-month exposure window—in the Methods section (page 43, lines 831-836), we have also mentioned the rationale: “Given that previous research emphasized adverse effects of maternal and in-utero exposures on early childhood health and episodes of El Niño and La Niña typically last nine to 12 months,^{3,7} we focused on examining the overall cumulative associations with ENSO exposure levels up to 12 months prior to the birth date of each child (covering mothers’ preconceptional and prenatal periods).” We aim to investigate the combined effect of the ENSO exposure levels.**

Moreover, to further justify the use of lag 0-12 months, in this revision, we have

conducted separate analyses for the preconception and pregnancy periods. In the updated Extended Data file, we have provided the cumulative results for lag 9-12 (preconceptional) months in Figure S5 and those for lag 0-8 (prenatal) months in Figure S6. It turns out that the shapes for the two separate periods are similar, and the effects for preconceptional months are slightly smaller. This is consistent with Figure 3 where effects are consistently found in the 0th-1st and 6th-12th months. The findings from separate analyses thus serve as another justification for combining the two periods in our main analysis. Note that, the separate analyses are provided as a sensitivity analysis. In the Methods section (pages 46-47, lines 900-906), we wrote the following:

“We firstly separated the analyses for assessing the impact of ENSO exposure during the mothers’ preconceptional and prenatal periods. Given the lack of exact information on maternal gestational age in the DHS, we assumed that each child birth was carried to a full 9-month term as previously estimated.^{8,9} We therefore assigned the ENSO exposure levels during the 9th-12th lagged months prior to birth as the ‘preconceptional’ exposure periods and the 0th-8th lagged months prior to birth as ‘prenatal’ exposure periods.”

c. Please clarify the reason for excluding neonatal mortality in stratified analysis.

Response: Thank you for reminding us of the potential issue. Previously it was simply excluded to save space. However, in this revision, we have added the results for neonatal mortality in stratified analysis (see the updated Extended Data Figure S2).

d. Multiple comparison: The authors used factor-specific Bonferroni-corrected p-values for the effect modification analysis. However, the authors had no consideration of multiple factors (e.g., sex, birth order, mother age).

Response: We really appreciate your comment on multiple comparisons. To further consider multiple factors, in this revision, we have adjusted the p-values (or critical values) by taking into account the total number of hypothesis tests, for a specific outcome and the chosen ENSO index. In the Methods section (page 45, lines 861-869, we wrote:

“For each outcome measure and a chosen ENSO index, the Benjamini-Hochberg adjustment was applied to account for multiple comparisons in examining the differences between subgroups in our effect modification analyses, and the cutoff for statistical significance was determined by the false discovery rate (FDR) < 0.05 .¹⁰ The Benjamini-Hochberg critical value is equal to $\frac{i}{m} \times Q$, where Q is set at 0.05 according to the FDR, m is the total number of hypothesis tests (in our case, $m = 25$), and i is the ranking of the p-value among the p-values of all the tests with a smaller number denoting a smaller p-value. When $i = 1$ (i.e., for the smallest p-value), it is equivalent to the Bonferroni adjustment.”

We adopted the Benjamini-Hochberg adjustment given the large number of tests we conducted for each outcome and chosen ENSO index. There were 25 hypothesis tests we conducted: 20 p-values (excluding the reference) from Figure 4, 1 p-value from Extended Data Table S4, and 4 p-values from Extended Data Table S5, for each outcome and chosen ENSO index. The adjusted statistical significance was marked by * in the figures and tables. If we use the more conservative Bonferroni adjustment, 12 out of the 47 rejected null hypotheses will not be rejected. That is to say, 35 out of the 47 null hypotheses are rejected under both the Benjamini-Hochberg and Bonferroni adjustments.

e. Country effects: Any specific reasons for controlling country effects instead of cluster effects?

Response: Thank you for bring up the cluster effects. Our main regressions control for country (fixed) effects, as specified in the Methods section (page 43, lines 827-829): “...all Cox models additionally adjusted for the country fixed-effects to explain potential differences across the countries (with Angola chosen as the reference category) similar to previous studies.^{9,11}” This is in line with Reviewer #1’s second major query in the previous round where the reviewer reminded us to consider country as a possible covariate.

To further incorporate your kind suggestion, we further consider two sensitivity analyses: 1) replace country fixed-effects by random effects; 2) further introduce the random effect for each DHS survey cluster code, which is constructed from the 2-character country code, 4-digit survey year, and 8-digit cluster identification number—this is the 14-character variable DHSID that uniquely identifies clusters across samples in our data. In the Methods section (page 47, lines 906-911), we wrote:

“Second, Cox proportional hazard models were repeatedly conducted to analyze the association with ENSO exposure by including random effects for different countries.¹² Third, the random effect for each survey cluster was also considered to adjust for cross-cluster differences in the DHS.¹³ For the second and third sensitivity analyses, Cox frailty models were applied with random intercepts to account for clustering effects.¹⁴”

As specified in the cited research,¹³ clusters were randomly selected from a list of census enumeration areas stratified by geographic region and by urban/rural area within each region in the DHS sample design, and households were then randomly selected from each cluster.

Then, as suggested by Extended Data Figure S7, the association patterns were generally consistent with our observations in the main models. This indicates that

clustering effects are not likely the drivers of our results.

f. Subgroup analyses by temperature and precipitation: As explained by the authors in the rebuttal, temperature and precipitation serve as mediators. Then how would authors interpret the ENSO estimates when stratified by mediators? Were they total effects of ENSO minus the effects through temperature/precipitation? Please be cautious here that the interpretation changed.

***Response:* We thank the reviewer for raising this concern. In fact, we consider temperature and precipitation both mediators and moderators, after checking the various settings in the literature. Note that, in Panels B and C of Figure 1, we have tried to understand how these meteorologic parameters could mediate the effect of ENSO levels on child mortality. Whereas mediation analysis was not conducted due to its incompatibility with our nonlinear models, we have included relevant discussions in lines 337-350 of page 16 under the Discussion section. To further investigate the potential moderating effects, we have conducted stratified analyses by temperature and precipitation. Admittedly, the relationships can be complex, and we have added a caution in our discussion of limitations (page 20, lines 427-430):**

“Fifth, we should also limit our interpretations of the results to the regions shown in Figure 1, and how the meteorologic parameters serve as both mediators and moderators requires further investigations.”

g. Classification of ENSO measures by 0.5 and 1: ENSO had associations with outcomes starting from values smaller than 0.5. Why did the authors still consider -0.5 to 0.5 as the reference group? How would that bias the estimates?

***Response:* Thank you for raising this concern. In this revision, we have added a justification for our selection of the interval [-0.5, 0.5] in the Exploratory Analyses**

section (pages 9-10, lines 196-199): “According to the United States National Oceanic and Atmospheric Administration (NOAA) operational definitions for El Niño and La Niña conditions, we assessed the health impacts of ENSO measures meeting or exceeding +/- 0.5°C, compared to ENSO-neutral status.” This suggests that our selection of the interval is in line with NOAA’s standard, enabling an easier transition from our result interpretations to policy implications.

Admittedly, the effects typically start to show up before our ENSO index reaching +0.5, which suggests that the risk in the reference group may actually be greater than zero, thereby leading to an underestimation of the difference between the high-exposure group and the reference group. With a truly “neutral” (i.e., no-risk-at-all) group, the difference in risk (and thus HR) should be larger, and hence the AF should also be larger. Therefore, our results are more conservative and may serve as lower bounds. In lines 208-210, we added a caution: “Note that, since there may also be risks in the ‘neutral’ conditions based on Figure 2, using this reference may underestimate the AF and thus, provide more conservative insights.”

h. Calculation of attributable fractions: The formula is for RR, not HR. The authors should provide theoretical reasons in the extended data, why such an equation is applicable.

Response: Thank you for pointing out this issue. In this revision, we have corrected the formula and then provided the justification for using HR in the formula (page 45, lines 875-877): “As deaths are usually considered rare events, HR and relative risk (RR) were assumed to be a reasonable approximation of each other in this study with caution.¹⁵⁻¹⁷” Note that, the theoretical reasons have been provided in the cited papers, e.g., the paper from the *Advances in Methodology & Statistics* provides a mathematical derivation showing the approximate equivalence of HR and RR when an event is rare.¹⁵ We remind readers that caution in interpretation is still needed.

5. Results

a. Table 1: What is the meaning of numbers in parentheses?

Response: Thank you for asking for clarity. In this revision, we have added notes under Table 1 to clarify that numbers in the parentheses are standard deviations when the numbers outside the parentheses are means, and are percentages when the numbers outside are counts. In the first column, we also have attached “n (%)” and “mean (SD)” to the corresponding variables to indicate the meanings of the numbers for each row (variable).

b. Table 1: The way of showing the mother age group is not clear. For example, is 20 years old included in the first group or second group?

Response: Thank you for asking for clarity. In this revision, to avoid confusion, we have used non-overlapping intervals to indicate the age groups. For example, 20 years old was not included in the first group, only the second group. We realized that it may not be clear to readers that the brackets “[]” and parentheses “()” in math distinguish inclusion or not. The revision is thus made.

Minor comments

1. Line 666: “(death at <12 months of age”, a “(“ is missing.

Response: Thank you for catching this! We have added the missing “)”.

2. Line 766: “Also, we assessed weather ...” should be “whether”?

Response: Thank you for catching this typo! We have corrected.

We sincerely hope that the concerns raised by our respected reviewers have been

properly addressed and that our revised paper is further considered by *Nature Communications*. We are grateful for this chance to further improve our paper, and we look forward to hearing more from the journal!

Reference

- 1 Higgins, J. P. T. & Thompson, S. G. Quantifying heterogeneity in a meta-analysis. *Statistics in Medicine* **21**, 1539-1558 (2002). <https://doi.org/10.1002/sim.1186>
- 2 Caleyachetty, R. *et al.* Tobacco use in pregnant women: analysis of data from Demographic and Health Surveys from 54 low-income and middle-income countries. *Lancet Glob Health* **2**, E513-E520 (2014). [https://doi.org/10.1016/S2214-109x\(14\)70283-9](https://doi.org/10.1016/S2214-109x(14)70283-9)
- 3 World Health Organization. 2023. *Public Health Situation Analysis: El Niño*. <https://www.who.int/publications/m/item/public-health-situation-analysis--el-ni-o> (accessed 4 August 2023).
- 4 Christian, L. M. Physiological reactivity to psychological stress in human pregnancy: Current knowledge and future directions. *Prog Neurobiol* **99**, 106-116 (2012). <https://doi.org/10.1016/j.pneurobio.2012.07.003>
- 5 Rylander, C., Odland, J. & Sandanger, T. M. Climate change and the potential effects on maternal and pregnancy outcomes: an assessment of the most vulnerable--the mother, fetus, and newborn child. *Global health action* **6**, 19538 (2013). <https://doi.org/10.3402/gha.v6i0.19538>
- 6 Heaney, A. K., Shaman, J. & Alexander, K. A. El Niño-Southern oscillation and under-5 diarrhea in Botswana. *Nature communications* **10**, 5798 (2019). <https://doi.org/10.1038/s41467-019-13584-6>
- 7 Heindel, J. J. *et al.* Developmental Origins of Health and Disease: Integrating Environmental Influences. *Endocrinology* **156**, 3416-3421 (2015). <https://doi.org/10.1210/en.2015-1394>
- 8 Liao, J. *et al.* Child Survival and Early Lifetime Exposures to Ambient Fine Particulate Matter in India: A Retrospective Cohort Study. *Environ Health*

- Perspect* **130**, 17009 (2022). <https://doi.org/10.1289/EHP8910>
- 9 Goyal, N., Karra, M. & Canning, D. Early-life exposure to ambient fine particulate air pollution and infant mortality: pooled evidence from 43 low- and middle-income countries. *International journal of epidemiology* **48**, 1125-1141 (2019). <https://doi.org/10.1093/ije/dyz090>
- 10 Lee, S. & Lee, D. K. What is the proper way to apply the multiple comparison test? *Korean Journal of Anesthesiology* **71**, 353-360 (2018). <https://doi.org/10.4097/kja.d.18.00242>
- 11 Odo, D. B. *et al.* Ambient air pollution and acute respiratory infection in children aged under 5 years living in 35 developing countries. *Environ Int* **159**, 107019 (2022). <https://doi.org/10.1016/j.envint.2021.107019>
- 12 Bickton, F. M. *et al.* Household air pollution and under-five mortality in sub-Saharan Africa: an analysis of 14 demographic and health surveys. *Environ Health Prev* **25** (2020). <https://doi.org/10.1186/s12199-020-00902-4>
- 13 Wang, P., Asare, E., Pitzer, V. E., Dubrow, R. & Chen, K. Associations between long-term drought and diarrhea among children under five in low- and middle-income countries. *Nature communications* **13** (2022). <https://doi.org/10.1038/s41467-022-31291-7>
- 14 Zhu, Y. *et al.* Socioeconomic disparity in mortality and the burden of cardiovascular disease: analysis of the Prospective Urban Rural Epidemiology (PURE)-China cohort study. *Lancet Public Health* **8**, e968-e977 (2023). [https://doi.org/10.1016/S2468-2667\(23\)00244-X](https://doi.org/10.1016/S2468-2667(23)00244-X)
- 15 Stare, J. & Boulch, D. Odds Ratio, Hazard Ratio and Relative Risk. *Advances in Methodology and Statistics* **13**, 59-67 (2016). <https://doi.org/10.51936/uwah2960>
- 16 Dadi, A. F., Miller, E. R. & Mwanri, L. Postnatal depression and its association with adverse infant health outcomes in low- and middle-income countries: a systematic review and meta-analysis. *Bmc Pregnancy Childb* **20** (2020). <https://doi.org/10.1186/s12884-020-03092-7>
- 17 Picetti, R. *et al.* Nitrate and nitrite contamination in drinking water and cancer

risk: A systematic review with meta-analysis. *Environ Res* **210** (2022).

<https://doi.org/10.1016/j.envres.2022.112988>

REVIEWERS' COMMENTS

Reviewer #1 (Remarks to the Author):

I was almost ready to sign off on the previous version of this paper so I don't want to belabor this report. Rereading the paper, however, I did spot a small number of textual issues that I would still like to see addressed, and also have one more substantial query. I'll deal with the substantial query first.

This concerns the multiple comparisons analysis. Reviewer 4 says "The authors used factor-specific Bonferroni-corrected p-values for the effect modification analysis" but requested further analysis of multiple factors, to which the authors responded by making some calculations using the Benjamini-Hochberg false discovery rate (FDR) procedure.

Well, maybe I just missed it or misinterpreted something in the original version of the paper, but a word search of the previous version of the paper for "Bonferroni" or "multiple comparisons" did not produce any result, so I don't know what reviewer 4 was referring to. My concern, however, is not whether I agree or disagree with this particular reviewer, but the clarity of the authors' response. Please remember, you're not just responding to the reviewers, but writing for the entire readership of the journal. And I'd prefer to omit the discussion of multiple comparisons altogether than to create a misleading impression for the readership.

This discussion impacts Figure 4 of the main paper and a couple of tables in the supplementary materials. The caption of Figure 4, however, doesn't make clear to which set of hypotheses the Benjamini-Hochberg procedure is being applied, or why the authors are even doing this. The results will be confusing to the reader because, in the end, only four of the results in Figure 4 get the all-important asterisk, which naturally raises the question of how to interpret all of the others.

I've a couple of comments on this. First, I don't think the standard FDR procedure is even available in this setting - the original proof of Benjamini-Hochberg required independence of the test statistics, and that doesn't seem applicable here, because when analyzing different covariates within a common set of regression analyses, the test statistics are dependent. There are various extensions of FDR that allow for dependence, but typically they yield results much closer to the Bonferroni procedure, which doesn't require independence, but which is also over-conservative in most situations.

But second, why bother with a multiple comparisons correction at all? Statisticians often distinguish between "exploratory" and "confirmatory" analyses, with the implication that one must be much more careful about precise statistical procedures when making confirmatory analyses. I think Figure 4 is exploratory - the authors are investigating which individual factors might be associated with a stronger El Nino effect, and the results should be interpreted in that way - to take one example from the body of Figure 4, it does indeed look as though "education" is a significant effect modifier, but no claim is being made that this is a more definitive effect than that for sex, birth order, or any of the other factors considered. I think the interpretation of Figure 4 is clear as it is, and does not need elaboration.

My minor points (inherited from the previous version):

l. 94, "many El Nino episodes have developed in 2023" (actually, the word "many" is new since l. 84 of the previous version) I am wondering what the authors meant by this. As I understand, there was an extended El Nino event during 2023, but the relevance of this to future events is not clear to me. Maybe a bigger concern is that with El Nino cycles continuing to take place against a background of climate change in general, there is a likelihood that the strongest El Nino events will be associated with even more extreme weather variables in the future. Maybe this is what the authors meant.

l. 228: the authors quote the combined HR of 1.39 (CI, 1.21 to 1.59) but when they use the phrase "robust to our primary results", I wondered which primary results they meant. The one

explicit comparison I can see is l. 149 where the HR for under-five mortality is quoted as 1.48 (1.43, 1.54) - is this the relevant comparison? The CI is quite a bit wider under the random effects analysis, which is easily understandable if there really is a wide country to country variation, but perhaps does not fit the description "robust".

l. 859/860: This is just a notational suggestion but it's quite common to put a "hat" (in French, circumflex accent) over the beta symbol in this sort of expression and this is quite easily added in Word (from the main display hit "Insert" then "Equation" then "Insert New Equation"; a box opens up; copy the beta symbol into that, then backspace so the cursor is back on top of the beta, then hit "Accent" in the menu bar; the "hat" symbol is one of many that are available). I would recommend that as a very small improvement to the notation.

However the authors respond to these points I am satisfied with this version and so not need to see any further revision.

Reviewer #1 (Remarks on code availability):

The code is much more extensive than the previous version of the paper (about 10 times as long). I didn't try to assess whether it works.

Reviewer #2 (Remarks to the Author):

I have reviewed the changes made to the manuscript based on feedback from all reviewers, and I am pleased to see the comprehensive efforts made to address the concerns raised. I commend the authors for their detailed response to the suggestions regarding clarifying statistical methods and adding the African co-author, which enhances the cultural and contextual depth of the study. The efforts to expand on the rationale for using specific statistical tests and including GitHub references for the computer code demonstrate a commitment to transparency and reproducibility. Moreover, I appreciate the addition of new analyses and discussions in response to my comments on the limitations concerning self-reported data and recall bias. Acknowledging these limitations and the subsequent analytical adjustments provide a more nuanced understanding of the research findings.

Incorporating interaction terms with environmental variables in response to feedback from other reviewers introduces a necessary complexity to the analysis, enriching the manuscript's scientific merit. Additionally, the thoughtful discussion of sample size representativeness and potential confounders further contributes to a more comprehensive and balanced view of the study's implications.

Given the substantive improvements made to the manuscript and the thorough consideration of all feedback provided, I have no further comments, and I look forward to its contribution to the field.

Reviewer #2 (Remarks on code availability):

The authors provide a README file with instructions for data preparation. The code itself includes detailed instructions at every step regarding variable selection and which plots, figures, or tables resulted from each code section. I was able to install the code; however, to run it, it would be required to use their dataset, which goes beyond the review process at this stage.

Reviewer #4 (Remarks to the Author):

I appreciate the efforts made by the authors. All my concerns have been addressed. I have no further comments.

Response to Reviewer Comments

We are extremely grateful for the insightful final comments from the anonymous reviewers and the chance to make some final changes in our manuscript “Maternal Preconceptional and Prenatal Exposure to El Niño Southern Oscillation Levels and Child Mortality: A Multi-country Study” (NCOMMS-23-11844B), based on these additional comments. Below, we have provided the point-by-point responses (in bold) to the comments (in plain text).

Reviewer #1 (Remarks to the Author):

I was almost ready to sign off on the previous version of this paper so I don't want to belabor this report. Rereading the paper, however, I did spot a small number of textual issues that I would still like to see addressed, and also have one more substantial query. I'll deal with the substantial query first.

***Response:* We really appreciate your careful final check of the paper. We have tried our best to address the final points raised below.**

This concerns the multiple comparisons analysis. Reviewer 4 says "The authors used factor-specific Bonferroni-corrected p-values for the effect modification analysis" but requested further analysis of multiple factors, to which the authors responded by making some calculations using the Benjamini-Hochberg false discovery rate (FDR) procedure.

Well, maybe I just missed it or misinterpreted something in the original version of the paper, but a word search of the previous version of the paper for "Bonferroni" or "multiple comparisons" did not produce any result, so I don't know what reviewer 4 was referring to. My concern, however, is not whether I agree or disagree with this particular reviewer, but the clarity of the authors' response. Please remember, you're not just responding to the reviewers, but writing for the entire readership of the journal.

And I'd prefer to omit the discussion of multiple comparisons altogether than to create a misleading impression for the readership.

***Response:* We agree with you that “multiple comparisons” were not a part of the original plan in the previous version of the paper. Given the concerns raised, we omit the discussion of multiple comparisons, and make clear to readers that we do not intend to identify a “more definitive” effect.**

This discussion impacts Figure 4 of the main paper and a couple of tables in the supplementary materials. The caption of Figure 4, however, doesn't make clear to which set of hypotheses the Benjamini-Hochberg procedure is being applied, or why the authors are even doing this. The results will be confusing to the reader because, in the end, only four of the results in Figure 4 get the all-important asterisk, which naturally raises the question of how to interpret all of the others.

***Response:* We have followed your kind suggestion to take the Benjamini-Hochberg procedure out of Figure 4, and Supplementary Figure S2 and Tables S4 and S5, given the potential misleading impression created for readers and the limitation of the adjustment you pointed out below.**

I've a couple of comments on this. First, I don't think the standard FDR procedure is even available in this setting - the original proof of Benjamini-Hochberg required independence of the test statistics, and that doesn't seem applicable here, because when analyzing different covariates within a common set of regression analyses, the test statistics are dependent. There are various extensions of FDR that allow for dependence, but typically they yield results much closer to the Bonferroni procedure, which doesn't require independence, but which is also over-conservative in most situations.

But second, why bother with a multiple comparisons correction at all? Statisticians often distinguish between "exploratory" and "confirmatory" analyses, with the

implication that one must be much more careful about precise statistical procedures when making confirmatory analyses. I think Figure 4 is exploratory - the authors are investigating which individual factors might be associated with a stronger El Nino effect, and the results should be interpreted in that way - to take one example from the body of Figure 4, it does indeed look as though "education" is a significant effect modifier, but no claim is being made that this is a more definitive effect than that for sex, birth order, or any of the other factors considered. I think the interpretation of Figure 4 is clear as it is, and does not need elaboration.

***Response:* Thank you for providing these two strong reasons against pursuing the multiple comparisons correction added in our last revision. We agree that Figure 4 and the relevant tables are intended to be exploratory instead of confirmatory, and thus, have removed the asterisks and associated notes from them.**

My minor points (inherited from the previous version):

l. 94, "many El Nino episodes have developed in 2023" (actually, the word "many" is new since l. 84 of the previous version) I am wondering what the authors meant by this. As I understand, there was an extended El Nino event during 2023, but the relevance of this to future events is not clear to me. Maybe a bigger concern is that with El Nino cycles continuing to take place against a background of climate change in general, there is a likelihood that the strongest El Nino events will be associated with even more extreme weather variables in the future. Maybe this is what the authors meant.

***Response:* We agree with you that the word "many" may be a bit confusing, and thus have removed this adjective. And yes, your understanding is correct. We have incorporated your clarification in the final revision.**

l. 228: the authors quote the combined HR of 1.39 (CI, 1.21 to 1.59) but when they use the phrase "robust to our primary results", I wondered which primary results they meant.

The one explicit comparison I can see is l. 149 where the HR for under-five mortality is quoted as 1.48 (1.43, 1.54) - is this the relevant comparison? The CI is quite a bit wider under the random effects analysis, which is easily understandable if there really is a wide country to country variation, but perhaps does not fit the description "robust".

***Response:* Yes, we are comparing 1.39 to 1.48. We agree that the CI is wider, and thus, have changed the word “robust” to “similar,” and remind the readers that the CI may not be quite as robust.**

l. 859/860: This is just a notational suggestion but it's quite common to put a "hat" (in French, circumflex accent) over the beta symbol in this sort of expression and this is quite easily added in Word (from the main display hit "Insert" then "Equation" then "Insert New Equation"; a box opens up; copy the beta symbol into that, then backspace so the cursor is back on top of the beta, then hit "Accent" in the menu bar; the "hat" symbol is one of many that are available). I would recommend that as a very small improvement to the notation.

***Response:* We have followed your suggestion to add the “hat” over beta symbol.**

However the authors respond to these points I am satisfied with this version and so not need to see any further revision.

***Response:* Thank you for the encouragement and wonderful final comments. We have decided to respond to all these points properly by following your suggestions. We hope that this paper is now ready to proceed!**

Reviewer #2 (Remarks to the Author):

I have reviewed the changes made to the manuscript based on feedback from all reviewers, and I am pleased to see the comprehensive efforts made to address the concerns raised. I commend the authors for their detailed response to the suggestions regarding clarifying statistical methods and adding the African co-author, which enhances the cultural and contextual depth of the study. The efforts to expand on the rationale for using specific statistical tests and including GitHub references for the computer code demonstrate a commitment to transparency and reproducibility.

Moreover, I appreciate the addition of new analyses and discussions in response to my comments on the limitations concerning self-reported data and recall bias. Acknowledging these limitations and the subsequent analytical adjustments provide a more nuanced understanding of the research findings.

Incorporating interaction terms with environmental variables in response to feedback from other reviewers introduces a necessary complexity to the analysis, enriching the manuscript's scientific merit. Additionally, the thoughtful discussion of sample size representativeness and potential confounders further contributes to a more comprehensive and balanced view of the study's implications.

Given the substantive improvements made to the manuscript and the thorough consideration of all feedback provided, I have no further comments, and I look forward to its contribution to the field.

Response: Thank you again for your confirmation! Your kind suggestions given earlier have helped us greatly in improving this paper. We indeed are with you in hoping to see that the work adds values to the field.

Reviewer #4 (Remarks to the Author):

I appreciate the efforts made by the authors. All my concerns have been addressed. I have no further comments.

Response: We would like to express our greatest gratitude for your helpful advice given before, and we are glad to see that our revisions are satisfactory.

We are grateful for the chance to give a final check of our paper and hope that this final version is ready to be published on *Nature Communications!*